# Talking-Head Generation in Practice

Zhicheng Zhang*
University of New South Wales
Canberra, Australian Capital Territory, Australia
zhicheng.zhang2@unsw.edu.au

Lei Wang*
Griffith University
Brisbane, Queensland, Australia
Data61/CSIRO
Canberra, Australian Capital Territory, Australia
l.wang4@griffith.edu.au

Yongsheng Gao†
Griffith University
Brisbane, Queensland, Australia
yongsheng.gao@griffith.edu.au

Yu Zhang†
University of New South Wales
Canberra, Australian Capital Territory, Australia
m.yuzhang@unsw.edu.au

## Abstract

Talking-head generation has progressed rapidly in recent years, driven by advances in vision, speech, and generative modeling. Yet despite this momentum, the field lacks a clear, consolidated understanding of how its research themes, datasets, and evaluation practices have evolved. To address this gap, we curate a comprehensive corpus of over 100 influential works and derive a coherent semantic taxonomy that reveals the main directions shaping the area, including speech-to-motion representation, style- and emotion-aware animation, and high-fidelity diffusion models. Building on this taxonomy, we present the first longitudinal analysis of datasets and metrics used in talking-head generation from 2021 to 2025. Our findings uncover distinct trends: increasing dependence on audio-visual and emotion-rich datasets, and a rapid rise in newly proposed evaluation metrics, especially those targeting expression naturalness, audio-visual synchronization, and landmark accuracy or driving-signal alignment. These patterns indicate a shift in the community's priorities from frame-level realism toward semantic alignment, temporal coherence, and perceptual quality. By unifying methodological structure with quantitative historical insights, this survey offers concrete guidance for developing future talking-head systems, choosing appropriate benchmarks, and designing meaningful evaluation protocols. We expect the work to serve as a central reference for advancing expressive, controllable, and perceptually aligned talking-head generation. Our appendix is available here.

## CCS Concepts

• **General and reference** → **Surveys and overviews**; General literature; • **Computing methodologies** → *Machine learning*; *Computer vision tasks*.

---

*Equal contribution (co-first authors).
†Corresponding authors.

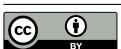

*WWW Companion '26, Dubai, United Arab Emirates*
© 2026 Copyright held by the owner/author(s).
ACM ISBN 979-8-4007-2308-7/2026/04
https://doi.org/10.1145/3774905.3794684

## Keywords

Talking-head generation; Facial animation; Audio-visual modeling; Diffusion-based generation; Dataset analysis; Evaluation protocols; Semantic taxonomy

**ACM Reference Format:**
Zhicheng Zhang, Lei Wang, Yongsheng Gao, and Yu Zhang. 2026. Talking-Head Generation in Practice. In *Companion Proceedings of the ACM Web Conference 2026 (WWW Companion '26), April 13–17, 2026, Dubai, United Arab Emirates.* ACM, New York, NY, USA, 28 pages. https://doi.org/10.1145/3774905.3794684

## 1 Introduction

Talking-head generation [6, 18, 21, 23, 33–35, 45, 47, 66], synthesizing realistic and expressive human head motions from audio, text, or driving signals, has rapidly transformed from a niche problem into a central capability for digital humans, virtual communication, education, entertainment, and accessibility technologies. Fueled by advances in deep generative modeling, modern systems can now produce high-fidelity facial motion [59, 63], nuanced emotional expression [20, 40], and identity-preserving appearance from minimal inputs [31, 58, 62]. The last five years, in particular, have seen an explosion of new approaches spanning motion-representation learning [12, 53], neural radiance fields (NeRFs) [14, 59], diffusion-based synthesis [11, 40], and multimodal control [56]. This rapid progress has pushed the field forward at an unprecedented pace, but it has also made it increasingly challenging to understand the broader landscape.

Despite this momentum, several fundamental questions remain unanswered. *What core problem formulations are driving current research? How are the latest models different in terms of representation, controllability, and expressiveness? Which datasets and evaluation protocols are shaping the community's standards, and how have they changed over time? As new models emphasize emotion, style, or semantic alignment, are existing metrics still adequate?* And most importantly, *what emerging trends reveal where the field is heading next?* The literature has grown too quickly and too heterogeneously for these questions to be answered through anecdotal understanding alone [32, 46, 60].

Existing surveys and tutorials [10, 19, 38, 41, 65] cover only narrow subsets of talking-head generation, *e.g.*, lip synchronization, animation frameworks, or specific model families, and thus do not

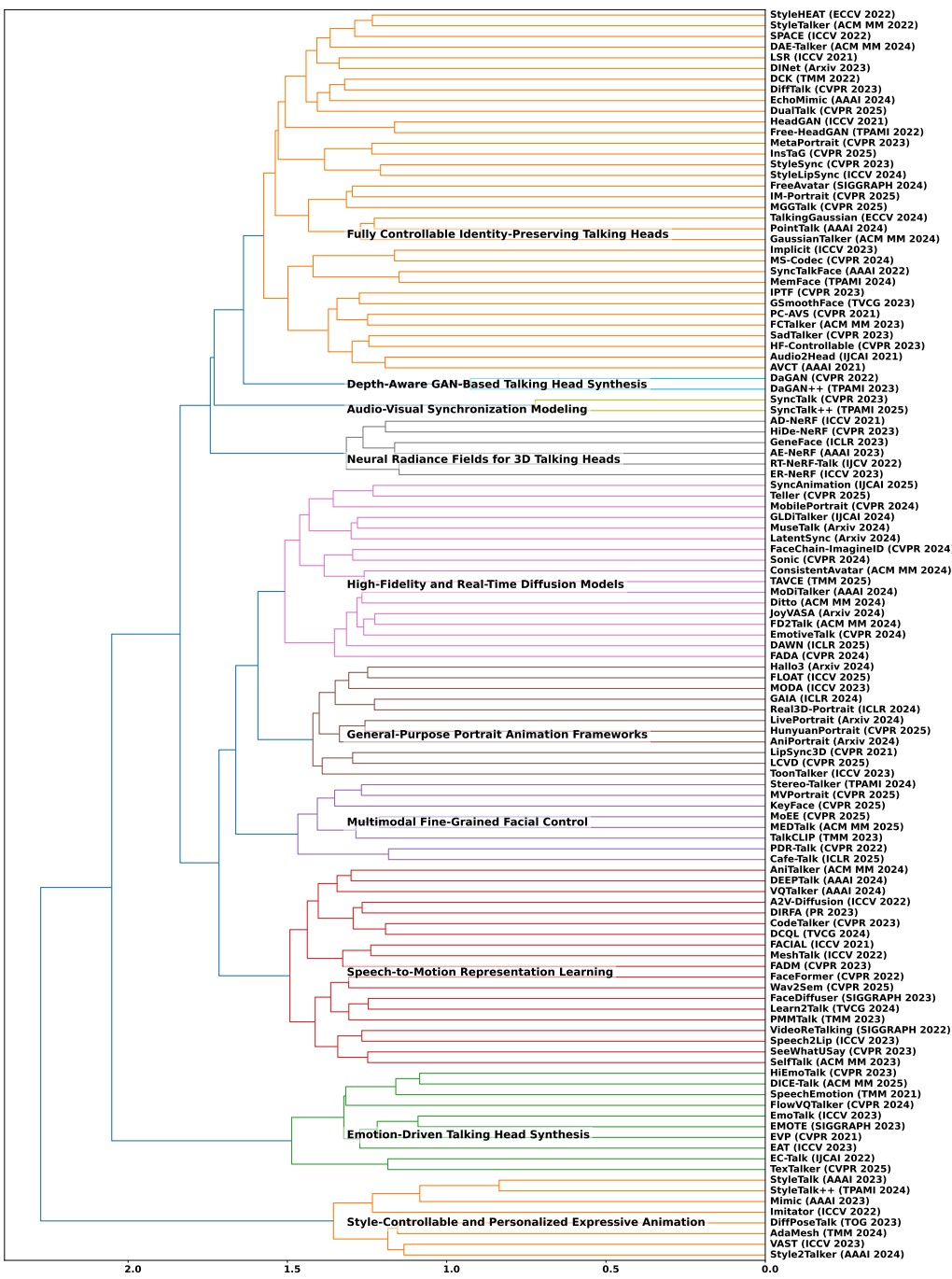

**Figure 1: Hierarchical semantic taxonomy of 117 representative talking-head generation methods from 2021-2025. The taxonomy shows ten major research directions that have naturally emerged from data-driven clustering of paper abstracts and titles, spanning foundational advances, such as speech-to-motion representation learning, NeRF-based 3D heads, and high-fidelity diffusion models, to higher-level themes in style control, emotion modeling, and multimodal fine-grained facial control. Adjacent clusters highlight areas of conceptual convergence, including the rise of expressive, personalized, and identity-preserving generation. The horizontal axis represents the linkage distance in hierarchical clustering, indicating the degree of semantic dissimilarity between groups of methods. Collectively, the taxonomy offers a structured map of the field's evolution, illustrating how modern talking-head research is diversifying from low-level realism toward semantic alignment, controllability, and fully generalizable portrait animation frameworks.**

capture the breadth, diversity, and evolution of the field. Meanwhile, experimental practices have shifted dramatically: datasets have expanded from celebrity video corpora [7, 30] to emotion-rich [25], high-resolution [64], and multi-view collections [50]; metrics have diversified from simple pixel-space measures [16, 51] to those evaluating semantic correctness [5, 13], audio-visual coherence [33, 57], and driving-signal alignment [53]. Yet there is currently no systematic, quantitative analysis showing how these choices have evolved and what they imply for future research.

To address these gaps, we curate a comprehensive corpus of more than 100 influential papers from 2021-2025 and analyze them through both semantic structure and experimental trends. Our goal is not only to map the methodological landscape but also to provide practical guidance for designing future systems, selecting appropriate benchmarks, and developing meaningful evaluation standards. The resulting survey aims to offer a clear, consolidated view of a rapidly evolving research area, and to serve as a foundation for the next generation of expressive, controllable, and perceptually aligned talking-head models. Our **main contributions** are threefold:

i. A unified taxonomy of talking-head generation, derived from a curated corpus of over 100 influential papers and revealing the major research directions that define the current landscape.

ii. The first longitudinal analysis of datasets and evaluation metrics (2021-2025), uncovering trends in dataset selection, benchmark evolution, and the community's shifting focus toward semantic, temporal, and perceptual evaluation.

iii. Actionable guidance for future research, highlighting experimental best practices, underexplored problem settings, and emerging opportunities for building more expressive, controllable, and reliable talking-head generation systems.

We review existing surveys and studies in Appendix; below, we present a unified perspective on this field from several aspects.

## 2 Framework for Unified Analysis

Appendix provides full details of the corpus curation process, semantic taxonomy methodology, and experiment extraction pipeline. Below we present our analytical framework and detailed analysis.

### 2.1 Analytical Framework

Our analytical framework is built on a curated corpus of 117 representative talking-head generation papers, selected through a multi-stage process designed to balance completeness and quality.

We begin with broad keyword-based retrieval from arXiv, then enrich each entry with metadata and code links. Peer-reviewed publications are automatically retained, while arXiv-only preprints are filtered using community impact signals, ensuring inclusion only when the work has demonstrable influence (*e.g.*, implementation popularity). A final semantic screening removes papers that fall outside the scope of explicit talking-head generation. This yields a corpus that captures the major model families (*e.g.*, GANs, NeRFs, diffusion models, transformers, *etc.*), the dominant control modalities (*e.g.*, audio-driven, text-driven, motion-driven, *etc.*), and the community's most visible and methodologically mature contributions. From this corpus, we extract all dataset and evaluation-metric information by parsing each paper's experiment section and processing it with a structured LLM-based extraction prompt. Dataset

names, metric names, and training/testing splits are normalized to unified identifiers, enabling consistent cross-paper comparison. These standardized elements form the basis for our longitudinal analyses of dataset usage and metric evolution.

To reveal the thematic structure of the field, we construct a hierarchical semantic taxonomy using TF-IDF [39] representations of titles and abstracts, followed by hierarchical agglomerative clustering with Ward's linkage. This method is chosen because it minimizes within-cluster variance and naturally exposes multi-level topic organization. The resulting dendrogram provides a visual and data-driven map of research themes, while cluster labels are refined using a large language model to ensure interpretability.

This pipeline (*e.g.*, representative corpus construction, structured experiment extraction, and hierarchical semantic mapping) forms a coherent framework that supports all subsequent analyses of trends, practices, and emerging directions in talking-head generation.

### 2.2 Semantic Taxonomy of Research Directions

Figure 1 presents the hierarchical semantic taxonomy derived from our corpus of 117 representative works, offering a view of how the talking-head generation community has organically structured itself over the past five years. It provides not just a static categorization, but a conceptual map that shows how the field has evolved, where current momentum is concentrated, and which thematic connections may shape the next generation of talking-head systems. Rather than imposing predefined categories, the taxonomy emerges directly from data-driven clustering, showing ten coherent research directions that together map the conceptual terrain of the field.

At the top of the taxonomy lie methodological backbones, such as speech-to-motion representation learning, NeRFs for 3D talking heads, and high-fidelity diffusion models. These clusters represent the generative foundations underpinning most recent advances. Their clear separation highlights how the community has diversified beyond traditional encoder-decoder pipelines: NeRF-based 3D heads emphasize spatial consistency and view-dependence; diffusion models drive improvements in realism and temporal stability; and representation-learning methods focus on robust, speaker-independent motion modeling. The size and density of these clusters indicate that they collectively power much of the field's technical momentum. A second set of clusters reflects emerging forms of control and expressiveness, including style-controllable and personalized animation, emotion-driven synthesis, multimodal fine-grained facial control, and audio-visual synchronization modeling. These areas illustrate a shift from lip-sync correctness toward richer communicative behaviors, emotional nuance, stylistic identity, controllable expressivity, and fine-grained semantic responsiveness. Their close spatial proximity in the taxonomy shows that many recent works operate at the intersection of these goals, blending speech cues, emotional priors, and multimodal signals to achieve more lifelike head motion.

Finally, two clusters, general-purpose portrait animation frameworks and fully controllable identity-preserving talking heads, capture systems that integrate multiple capabilities into unified frameworks. These works often pull techniques from several upstream clusters, reflecting a trend toward versatile pipelines that

support reenactment, instruction-based generation, long-term identity preservation, and user-controllable behavior. The prominent position of these clusters suggests that the community increasingly values practical, deployable solutions that can generalize across identities, styles, and input modalities.

> The taxonomy illustrates three key insights. First, the field has become structurally multidimensional: progress is no longer driven by a single innovation path, but by parallel advances across representation, controllability, realism, and user alignment. Second, there is a convergence toward models that reconcile expressiveness with reliability, seen in the adjacency of emotional, stylistic, and identity-preserving clusters. Third, the boundaries between categories are porous: many recent systems sit at the intersections, indicating a maturation of the field toward holistic, multimodal, and semantically grounded talking-head generation.

## 2.3 Longitudinal Trends in Dataset Usage

Figure 2 summarizes the longitudinal trends in dataset selection, grouped into four major categories, face image datasets, audio datasets, audio-visual datasets, and emotion-rich datasets, while distinguishing training and testing usage. The figure exposes a clear transition in the data foundations of talking-head research, reflecting the community's expanding requirements for expressiveness, multimodal alignment, and performance generalization.

In the early years, the landscape is dominated by audio-visual video datasets, particularly VoxCeleb1/2 [7, 30], LRS2/3 [1, 2], and GRID [9]. These datasets provided large-scale, speaker-diverse audio-visual pairs, making them ideal for traditional lip-sync and audio-driven head-motion models. Their consistent presence across both training and testing indicates their role as de facto benchmarks during this period. However, their relative dominance gradually declines as more specialized datasets begin to enter the ecosystem. From 2023 onward, we observe a rising dependence on high-resolution and multi-view portrait datasets, as seen in the growing use of CelebV-HQ [67], HDTF [64], and other sources. This trend mirrors the shift toward 3D-aware and diffusion-based models (see Figure 1), which require richer spatial variation, higher fidelity appearances, and more consistent lighting conditions to train view-consistent head renderers and NeRF-style representations.

A particularly notable trend is the steady increase in emotion datasets used for both training and evaluation. Starting from almost negligible usage in 2021, emotion datasets, such as CREMA-D [4] and RAVDESS [25], become substantially more common in later years, especially by 2024-2025. This growth directly reflects the community's expanding interest in expressive and emotionally aligned talking-head generation, where models should move beyond neutral lip movement to capture affective nuance, speaking style, and conversational authenticity. Meanwhile, audio-only datasets maintain consistent but modest usage, primarily serving as supplementary resources when large-scale or domain-matched speech is required. Their secondary role underscores that talking-head generation fundamentally relies on synchronized audio-visual data or emotion-annotated corpora, rather than speech alone.

Across all categories, the increasing diversity of training-testing splits highlights a broader methodological maturation: models are

tested on data distributions increasingly different from those they were trained on. This reflects growing emphasis on generalization, robustness, and evaluation fairness, central concerns as talking-head technologies move from controlled laboratory conditions to real-world applications.

> This shows an evolution from early reliance on generic, large-scale audio-visual datasets toward high-resolution, emotion-rich, multi-view, and semantically annotated datasets. This shift demonstrates that the field is no longer satisfied with generating visually plausible motion, but is now targeting identity robustness, expressive fidelity, and semantically coherent behavior, all of which demand more diverse and specialized data foundations.

## 2.4 Evolution of Evaluation Metrics

**Evaluation metric landscape.** Figure 3 presents a longitudinal analysis of evaluation metrics used in talking-head generation, organized into five core metric families: visual quality, audio-visual synchronization, geometry and landmark accuracy, expression naturalness, and driving-signal alignment. The figure shows a notable shift in how the community measures progress, reflecting deeper changes in research priorities and modeling capabilities.

Early in the timeline, research is dominated by visual-quality metrics, $e.g.$, SSIM [51], PSNR [16], LPIPS [43], and FID [15]. These metrics, inherited from general video synthesis, were sufficient when the primary challenge was achieving realistic textures, stable frames, and identity preservation. However, their relative decline after 2023 indicates a growing consensus that frame-level similarity is no longer the main bottleneck for state-of-the-art systems.

Starting in 2023, we observe a steady rise in audio-visual synchronization metrics ($e.g.$, LSE-C/D [8], Sync-C/D [33]), reflecting renewed attention to temporal coherence and phoneme-viseme alignment as models achieve higher realism but still struggle with precise timing. This trend underscores the community's recognition that perceptual synchronization is essential for speech-driven animation and profoundly impacts user trust and perceived naturalness. Even more striking is the rapid growth of geometry and landmark accuracy metrics and expression naturalness metrics. Metrics such as LMD, F-LMD, emotion accuracy, and smoothness [24, 29, 33, 42, 54] become increasingly common after 2023, peaking around 2024-2025. This shift shows a deeper transition in the field: from surface-level realism to semantic correctness, evaluating whether the generated facial motion matches linguistic cues, emotional intent, and human expressive norms. These metrics reflect the rising influence of emotionally aware and style-controlled models, which require more nuanced evaluation beyond pixel fidelity.

Driving alignment metrics ($e.g.$, APD, AKD, ARD) [18, 22, 61], see meaningful adoption closer to 2025. Their growth suggests an emerging focus on controllability, where the goal is not merely to produce realistic motion but to ensure that user-specified control signals, motion trajectories, landmarks, expression codes, are faithfully reproduced. These metrics are critical for next-generation systems emphasizing fine-grained, user-driven manipulation.

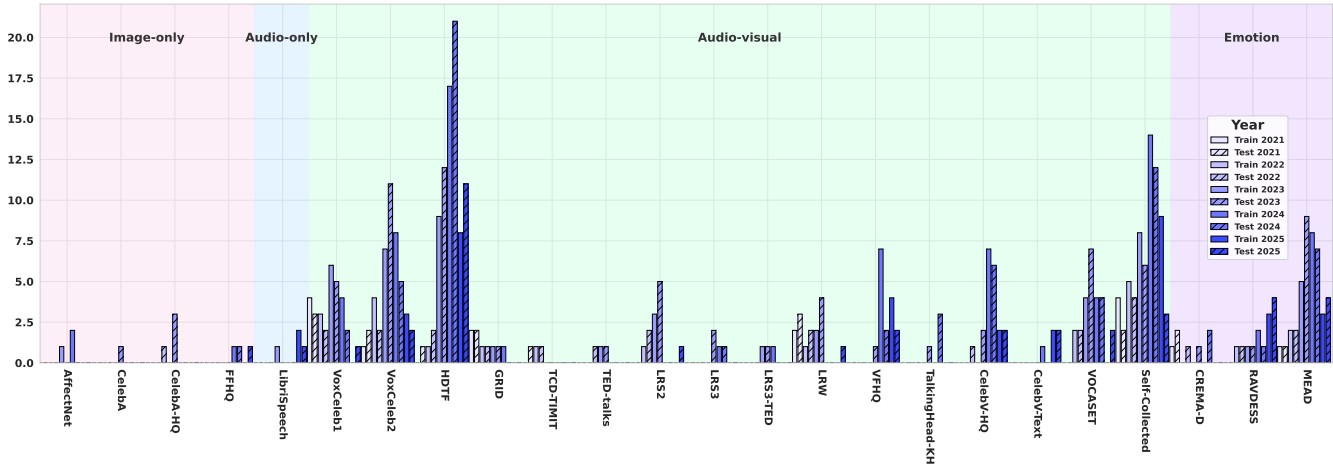

**Figure 2: Longitudinal trends in dataset usage. Datasets are grouped into four categories, face images, audio-only speech datasets, audio-visual datasets, and emotion-rich datasets, separately tracked for training and testing. The vertical axis represents the number of papers that use the corresponding dataset in a given year. The figure highlights a shift from early reliance on large-scale audio-visual benchmarks (*e.g.*, VoxCeleb, LRS, *etc.*) toward growing adoption of high-resolution portrait, multi-view, and emotion-annotated datasets (*e.g.*, HDTF, MEAD, RAVDESS, *etc.*). This evolution reflects the field's increasing emphasis on expressiveness, semantic alignment, and generalization beyond standard lip-sync benchmarks.**

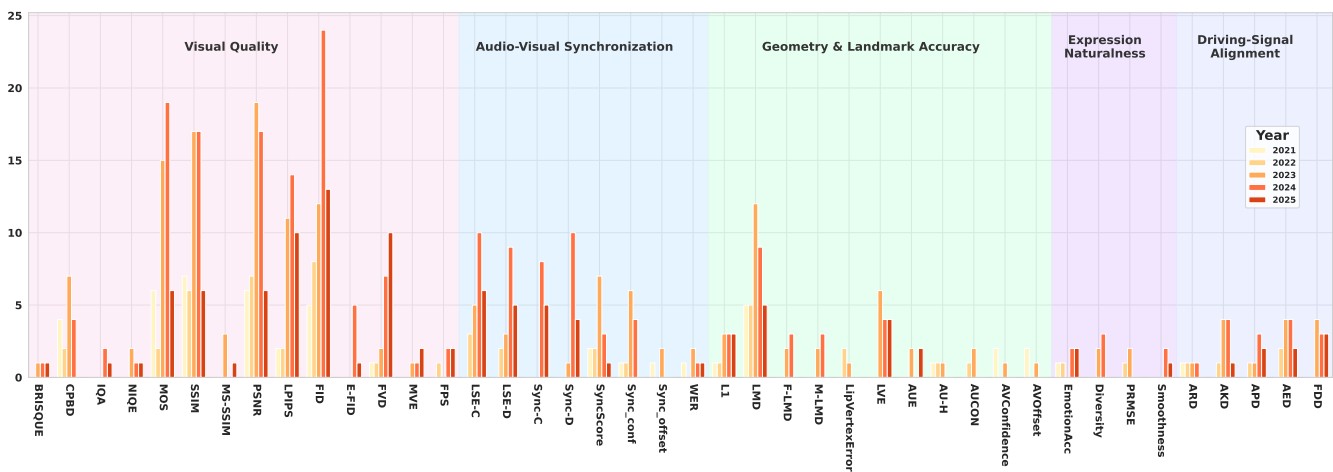

**Figure 3: Longitudinal evolution of evaluation metrics used. Metrics are grouped into five functional categories: visual quality, audio-visual synchronization, geometry/landmark accuracy, expression naturalness, and driving-signal alignment. The vertical axis represents the number of papers that use the corresponding evaluation metric in a given year. The trends show a clear shift from early reliance on pixel-level fidelity measures toward richer, semantically oriented evaluation, including synchronization accuracy, emotional expressiveness, and controllability. This progression highlights the community's move toward assessing temporal coherence, semantic alignment, and expressive behavior rather than surface-level realism alone.**

Our analysis shows a clear trajectory: the field is maturing from evaluating how real the frames look to how semantically correct, expressive, and controllable the generated motion is. The increasing diversity of metrics also reflects an expanding understanding of talking-head generation as a multimodal communication task, not just a visual synthesis problem. This evolution highlights the need for more unified, perceptually grounded metrics and suggests that future progress will depend on evaluation protocols capable of capturing high-level behaviors, *e.g.*, emotional resonance, conversational authenticity, and user-intent alignment.

**Shifts in metric priorities over time.** Figure 4 provides a two-part view of how evaluation practices in talking-head generation have evolved across 117 papers. Figure 4a visualizes the usage frequency of the five major metric families each year, while Figure 4b shows how newly introduced metrics are distributed across these same categories. They show not only what the community measures, but how its priorities and innovation efforts have shifted as models have progressed from basic lip-sync generation to expressive, controllable, and multimodal talking-head systems.

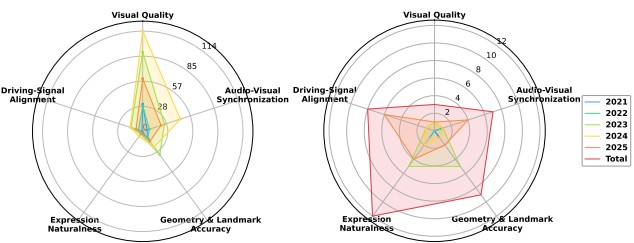

**(a) Metric usage trends.**     **(b) New metric distribution.**

**Figure 4: Evolution of evaluation metrics. Both subfigures share the same legend. (a) Radar visualization of metric usage across five major dimensions, showing the field's early dependence on visual-quality measures and its gradual shift toward audio-visual synchronization and geometric accuracy as models pursue greater temporal coherence and semantic correctness. (b) Distribution of newly proposed evaluation metrics over time, highlighting a clear rise in measures targeting expression naturalness, driving-signal alignment, and fine-grained motion accuracy.**

The left radar plot (Figure 4a) shows that visual-quality metrics (*e.g.*, LPIPS, SSIM, PSNR) dominated early years, reflecting a stage where improving sharpness, identity fidelity, and artifact reduction was the foremost challenge. Their relative share decreases over time, not because they disappear, but because progress in generative backbones (GANs → NeRFs → diffusion models) has reduced the centrality of pixel-level fidelity as the primary bottleneck. Meanwhile, the remaining four dimensions: audio-visual synchronization, geometry and landmark accuracy, expression naturalness, and driving-signal alignment, all expand steadily, indicating a broadened understanding of what constitutes good talking-head generation. Synchronization metrics grow notably after 2023, aligned with audio-aware diffusion models and transformer-based speech, motion encoders that require precise temporal consistency. Geometry-related metrics increase in step with the rise of 3D-aware and NeRF-based heads, where spatial correctness and stable structure matter as much as appearance.

The right radar plot (Figure 4b) shows the direction of metric innovation. Newly proposed metrics concentrate not in visual quality, where mature measures already exist, but in expression naturalness, landmark/geometry accuracy, and driving-signal alignment. This indicates that researchers are actively developing tools to evaluate behavioral and semantic performance: emotional expressiveness, physically plausible motion, responsiveness to control signals, and fine-grained alignment with speech or driving inputs. Synchronization-related metrics also appear among new proposals, reflecting a renewed focus on perceptual audio-motion coherence.

> The combined evidence from Figures 4a and 4b demonstrates a clear maturation of evaluation methodology. The field has moved beyond asking "Does the face look realistic?" toward "Does the face behave realistically, respond correctly, and communicate meaningfully?", a shift that mirrors the trajectory of modern talking-head research toward expressiveness, controllability, multimodal grounding, and human-aligned communication.

## 3 Findings and Emerging Trends

The analyses presented above collectively show a field that is not only expanding in capability but also maturing in scope and ambition. In this section, we synthesize these findings across three dimensions, models, datasets, and metrics, to highlight emerging trends, persistent challenges, and future opportunities.

### 3.1 Evolution of Model Paradigms

The semantic taxonomy in Figure 1 illustrates how the field has diversified into multiple coherent yet interconnected research directions. Early audio-driven models, which focused primarily on phoneme-viseme alignment and frame-level realism, have now been complemented, and in some cases surpassed, by multimodal and highly expressive approaches.

**Foundational modeling innovations.** Three clusters, speech-to-motion representation learning, NeRF-based 3D heads, and high-fidelity diffusion models, represent the core methodological engines driving recent progress. Representation-learning methods emphasize generalizable motion priors, NeRF-based systems elevate spatial realism and view consistency, and diffusion models substantially boost visual fidelity and temporal stability.

**Shift toward expressiveness and alignment.** Clusters related to emotion-driven synthesis, style control, and fine-grained multimodal manipulation demonstrate a clear pivot from lip-sync correctness to richer communicative behavior. Models introduced from 2023 onward increasingly incorporate emotion cues, prosody, semantic context, or user instructions, resulting in head motions that exhibit personality, style, and narrative coherence.

**Toward unified & controllable systems.** The emergence of general-purpose animation frameworks and fully controllable identity-preserving systems suggests a trend toward integrated solutions capable of reenactment, audio-driven generation, style modulation, and semantic alignment within a single architecture. These systems reflect the field's ambition to support real-world deployment, where controllability, rather than raw realism, is decisive.

> These model trends show a clear trajectory: talking-head generation is evolving from narrowly defined, modality-specific synthesis tasks into a broader paradigm of expressive, controllable, and semantically grounded digital human communication.

### 3.2 Transformation of Dataset Foundations

Figure 2 shows a parallel evolution in dataset usage. Early models depended heavily on large-scale audio-visual datasets such as VoxCeleb. These datasets enabled progress in lip synchronization and basic head motion generation but provided limited coverage of emotion, style, or high-frequency appearance details.

**Rise of high-fidelity and multi-view datasets.** As NeRF-based and diffusion-based models gained popularity, the reliance on high-resolution and multi-view datasets increased. Resources such as CelebV-HQ, HDTF, and other high-quality portrait collections provide the spatial richness needed for geometric consistency and fine-grained appearance modeling.

**Growing importance of emotion and expressive datasets.** One of the strongest trends after 2023 is the increased use of emotion-rich and affect-annotated datasets (*e.g.*, RAVDESS, MEAD,

and self-collected). These datasets support the modeling of expressive cues, an area that traditional lip-sync corpora cannot cover. Their rising adoption also reflects the community's recognition that expressive realism matters as much as pixel-level realism.

**Training-testing divergence as a mark of maturity.** As the field matured, models increasingly trained on specialized datasets while testing on diverse or unseen distributions. This shift indicates a push toward robustness, generalization, and fair benchmarking, all of which are essential for real-world applications.

> Dataset evolution mirrors model evolution: to generate expressive, controllable talking heads, the field is gradually adopting datasets that are richer in emotion, higher in resolution, and more diverse in viewing conditions.

### 3.3 Maturation of Evaluation Practices

Figures 2-4 collectively show a decisive evolution in how talking-head generation is evaluated. The shift from traditional visual metrics to semantically grounded measures aligns with the field's conceptual transformation.

**Pixel-level visual metrics.** SSIM, PSNR, and FID dominated early evaluation but gradually lost prominence as models reached high visual fidelity. These metrics were never sufficient to capture temporal coherence, emotional intent, or semantic correctness, and their decline reflects increased awareness of these limitations.

**Rise of audio-visual synchronization metrics.** Metrics such as LSE-C/D and SyncScore surged beginning in 2023, corresponding with growing emphasis on natural timing, speech-driven dynamics, and prosody-aware motion. Synchronization is now seen as a key determinant of perceived naturalness and trustworthiness.

**Geometry and landmark accuracy.** With the proliferation of 3D-aware and deformation-based models, landmark- and geometry-based metrics (*e.g.*, LMD, F-LMD) have become essential for evaluating structural correctness. Their adoption signals that models are now judged not only by appearance but by their ability to maintain identity, pose accuracy, and geometric consistency.

**Emergence of expression and driving-alignment metrics.** The steep rise in expression naturalness and driving-alignment metrics after 2023 is perhaps the most important trend. As models incorporate emotion, style, and multimodal control, evaluation should consider whether generated motion reflects intended affect, follows control signals, and behaves consistently over time.

> This metric evolution reflects a deeper philosophical shift: talking-head generation is no longer evaluated solely as a graphics problem, but increasingly as a communication problem, where semantics, timing, and expressiveness matter as much as pixel quality.

### 3.4 Practical Guidelines

Below we provide some practical guidelines.

We recommend adopting a compact, standardized metric suite that balances perceptual, temporal, geometric, and controllability signals. For perceptual quality report LPIPS and FID (or E-FID for expressive/affective domains) together with a frame-level visual fidelity measure (*e.g.*, PSNR). For synchronization report an audio-visual sync metric such as LSE-C/LSE-D or SyncScore and include

Sync-offset statistics. For geometry and identity preservation report landmark/vertex errors (LMD or LipVertexError) and an identity similarity score (*e.g.*, ArcFace). For expression and affect fidelity report AU consistency or emotion-classification accuracy on emotion-annotated test sets. For controllability report driver-alignment measures (AKD/APD/ARD) and a smoothness/stability metric (*e.g.*, velocity/acceleration MSE). Automatic evaluation should be complemented by human perceptual judgments, *e.g.*, run a small MOS study with at least 30 independent raters per method and report mean MOS with 95% confidence intervals. Compute metrics per-video and aggregate results using medians and interquartile ranges (and means where appropriate), and release evaluation scripts and exact hyperparameters to ensure reproducibility.

For test splits we suggest a multi-faceted protocol to stress generalization, expressiveness, and controllability. Use (i) an IID holdout by withholding 20% of speakers from the primary training corpus to measure in-distribution performance; (ii) a cross-dataset split that evaluates models on a high-fidelity portrait dataset (*e.g.*, CelebV-HQ or HDTF) unseen during training; (iii) an emotion/expressiveness split using emotion-annotated corpora (*e.g.*, CREMA-D, RAVDESS) to measure affective fidelity; and (iv) a low-shot personalization and novel-view split (1 - 5 frames for personalization and a multi-view/novel-view dataset such as HDTF) to evaluate identity stability and view consistency. For each split report per-condition statistics (gender, age, lighting) where available and pair automatic metrics with the MOS study described above. Finally, publish the exact split definitions, sampling seeds, and all evaluation code to enable fair, reproducible comparison.

### 3.5 Future Outlook

The evolution of talking-head generation from 2021 to 2025 reveals a field transitioning from narrow, task-specific pipelines to a broader paradigm of expressive, controllable, and semantically aligned digital human communication. By integrating insights from our taxonomy, dataset analysis, and evaluation trends, a coherent picture emerges of how the community has expanded its technical foundations and redefined its ambitions (see also Appendix).

From a modeling perspective, the field has moved decisively beyond traditional 2D appearance-based synthesis. Recent systems increasingly incorporate 3D-aware representations, multimodal encoders, and diffusion-based refinement modules that jointly enhance geometric consistency, expressive motion, and temporal stability. This progression reflects a shift from focusing on lip-sync realism alone to modeling communicative behavior more holistically, including emotion, prosody, speaking style, and semantic cues. A growing emphasis on controllability further underscores this shift. Modern architectures aim not only to generate photorealistic faces but also to expose interpretable control interfaces through landmarks, expression codes, style tokens, or textual instructions, enabling fine-grained and user-driven animation.

These changes in modeling practice have been accompanied by equally notable shifts in dataset usage. Early work relied heavily on large audio-visual corpora such as VoxCeleb, which remain valuable but are insufficient for capturing emotional nuance or high-quality geometric detail. The field now increasingly depends on high-resolution, multi-view, and emotion-rich datasets, which

offer the diversity and expressive coverage needed for training models that mimic natural conversational behavior. The trend toward training on specialized datasets while testing on diverse or unseen sets also signals a maturing understanding of generalization and robustness, which are essential for real-world deployment.

Evaluation practices have matured in parallel. Traditional visual-fidelity metrics, *e.g.*, SSIM, PSNR, or FID, have gradually lost dominance as researchers recognize their limitations in capturing temporal coherence or expressive realism. Synchronization measures, geometric and landmark accuracy metrics, and expression-oriented criteria have become central to evaluation, reflecting an understanding that speaking heads must be assessed not solely by their appearance but by the behavioral correctness and perceptual plausibility of their motion. This shift from pixel-level to semantic and perceptual evaluation highlights a broader reorientation of the field toward modeling communication rather than visual frames alone.

These developments point toward several promising opportunities. Semantic grounding and conversational consistency remain underexplored, and deeper integration of linguistic context could yield more coherent and contextually appropriate motion. Personalization at extremely low data cost is another challenge, as current approaches still require substantial footage to capture identity and expressive priors. Ethical considerations, including misuse prevention, watermarking, consent mechanisms, and bias mitigation, will become increasingly important as realism continues to improve. Finally, a longer-term frontier lies in unified generative humans that integrate head motion, gaze behavior, emotional state, and upper-body gesturing into a single cohesive model.

> Talking-head generation is entering a new phase where semantic correctness, expressive fidelity, controllability, and reliability are becoming as important as visual realism. The field's trajectory suggests that future progress will depend on jointly advancing model architectures, dataset design, evaluation protocols, and ethical frameworks. These directions offer practical guidance for researchers and practitioners seeking to build the next generation of expressive, trustworthy, and context-aware digital humans.

## 4 Ethics, Limitations, and Conclusion

**Ethical considerations.** Talking-head generation technologies carry significant ethical implications due to their potential for both beneficial and harmful use. While recent advances enable more natural, expressive, and controllable digital humans, the same capabilities can be misappropriated for impersonation, misinformation, harassment, and other malicious applications [3, 26, 27, 36, 49, 52, 68].

Many datasets used in this field contain real individuals whose images and voices may not have been collected with explicit consent for synthetic content generation, raising concerns about privacy, consent, and downstream data reuse.

In addition, demographic imbalances in commonly used training corpora may amplify biases in identity preservation, expressive fidelity, and emotional rendering, leading to unequal performance across gender, age, or ethnic groups [28, 48, 55]. To mitigate such risks, researchers should adopt responsible data handling practices, use consent-aware or licensed datasets whenever possible, report

demographic composition transparently, and consider integrating watermarking or provenance signals into generative pipelines.

> We recommend that future work include explicit discussions of intended use, potential harms, and safety constraints, and that evaluations incorporate robustness and misuse-resilience criteria. Addressing these ethical and societal issues is essential for ensuring that advances in talking-head generation contribute positively to human communication rather than eroding trust or enabling deceptive media [17, 37, 44].

**Limitations.** Although the analysis in this survey draws on a large and diverse corpus of 117 papers, several limitations may influence the completeness and generality of our findings. First, the paper selection is based on keyword-driven searches and public availability, which may bias the corpus toward popular research directions. Second, while our taxonomy construction combines TF-IDF clustering with LLM-assisted labeling, both steps introduce methodological sensitivity: clustering outcomes depend on vocabulary choices and similarity thresholds, and LLM summaries may reflect prompt phrasing or model priors. Third, the extraction of dataset and metric usage is subject to inconsistencies in how papers report their experimental setups, and despite manual verification, occasional omissions or misclassifications may remain. Fourth, our longitudinal trends reflect the publication distribution of 2021 - 2025 and should not be interpreted as causal evidence of methodological superiority. Finally, because our survey focuses on publicly documented methods, it cannot fully capture proprietary datasets, unpublished internal benchmarks, or emergent industrial practices. These limitations do not undermine the core insights but highlight the need for continued community curation, open artifacts, and standardized reporting protocols.

**Conclusion.** Our findings reveal a field that has rapidly progressed from frame-level lip-sync synthesis toward expressive, controllable, and semantically grounded head motion generation. Modern systems increasingly use 3D-aware representations, diffusion models, and multimodal control signals to achieve higher fidelity, stronger identity preservation, and more natural communicative behavior. In parallel, we observe a clear shift toward richer and more diverse datasets, including high-resolution, multi-view, and emotion-annotated corpora, and toward evaluation metrics that emphasize perceptual realism, temporal coherence, geometric correctness, and expressive fidelity. These developments reflect a growing recognition that talking-head generation should be assessed not only by visual similarity but also by its alignment with linguistic, emotional, and user-provided cues. The trajectory of the field points toward digital humans that are more trustworthy, expressive, and contextually aligned. We hope that the taxonomy, analyses, and insights presented here provide a useful foundation for future research, guiding the development of more controllable, perceptually aligned, and ethically responsible talking-head generation systems.

## Acknowledgments

Zhicheng Zhang is supported by the UNSW University International Postgraduate Award (UIPA). This research was conducted under the supervision of Lei Wang.

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

## A  Related Work

Research on talking-head generation spans multiple communities in computer vision, speech processing, computer graphics, and multimedia. Prior reviews and tutorials have discussed selective aspects of this area, yet none provide a comprehensive, quantitatively grounded analysis of models, datasets, and evaluation practices over time. In this section, we briefly summarize representative surveys most relevant to our topic and describe how our work differs.

**Surveys on talking-head generation and facial animation.** A number of surveys and tutorials have examined subdomains of talking-head generation, including facial animation, lip synchronization, or multimodal speech-driven motion. For example, early reviews focused on traditional graphics pipelines and parametric animation models, emphasizing blendshape manipulation and rule-based speech-to-lip generation. More recent surveys have outlined high-level deep-learning pipelines for face reenactment or lip-sync generation, summarizing canonical architectures such as encoder-decoder models, audio-to-landmark regressors, and GAN-based video synthesis [18, 35, 47, 65] . While these efforts provide valuable overviews, they share key limitations: (i) Narrow scope: Existing surveys typically cover only one subtask, such as lip-sync synthesis or face reenactment, without mapping the broader landscape of expressive talking-head generation. (ii) Static categorization: Prior taxonomies rely on manually defined categories that may not reflect the rapid diversification of recent approaches (*e.g.*, diffusion, NeRF-based heads, style or emotion control, instruction-driven generation). (iii) Limited temporal perspective: None of the existing surveys quantitatively evaluate how research focus, datasets, or metrics have evolved over time.

We provide the first comprehensive and semantically grounded taxonomy built from more than 100 representative papers, showing not only what categories exist but how the field has naturally organized itself. Unlike previous surveys, our taxonomy is derived from the literature rather than imposed upon it, offering a more faithful and up-to-date understanding of the major directions in talking-head generation.

**Surveys on deepfake synthesis, video manipulation, and portrait animation.** Another line of surveys addresses deepfake generation, video manipulation, or portrait animation more broadly. These works typically classify technologies for face swapping, reenactment, audio-driven animation, expression editing, or identity transfer. They discuss safety, ethics, and detection pipelines, reflecting their focus on manipulation robustness and security implications [10, 19, 69, 101, 150, 184]. However, these reviews differ from our interests in several ways: (i) Broader but shallower: Although they cover a wider range of facial manipulation techniques, they do not specialize in talking-head generation and thus provide limited depth on motion modeling, audio-visual synchronization, or fine-grained controllability. (ii) Lack of experimental analytics: Existing deepfake-oriented surveys rarely analyze dataset usage, metric evolution, or long-term trends: elements that are crucial for understanding research maturity and future directions in talking-head synthesis. (iii) Limited coverage of recent modalities: The rise of diffusion-based generators, NeRF-based 3D talking heads, and multimodal alignment methods is either minimally covered or absent due to their recency.

Our survey focuses specifically on talking-head generation, enabling deep analysis of its unique challenges, such as semantic drive signal alignment, emotional expressiveness, temporal coherence, and identity preservation. Moreover, we go beyond architectural overviews to provide a historical, data-supported analysis of how evaluation and datasets have shifted in the last five years.

**Surveys on audio-visual learning, speech-driven animation, and embodied communication.** Surveys in audio-visual machine learning often discuss cross-modal representation learning, speech-driven body or head animation, and multimodal affective computing [38, 41, 74, 182, 196, 233, 238, 245]. These works contribute insights into motion representations and human communication modeling but generally have different objectives: (i) Focus on perception, not generation: Many emphasize recognition, understanding, or representation learning rather than synthesis. (ii) Lack of synthesis-oriented taxonomies: These surveys do not categorize generation models based on animation control, identity preservation, or generative backbones. (iii) No study of metrics or benchmark dynamics: Because their emphasis lies in multimodal understanding, they do not track the evolution of evaluation protocols used for generative tasks.

While benefiting from insights in multimodal communication, our survey is centered on generation, offering a systematic comparison of synthesis approaches, control signals, and generative architectures. Furthermore, we explicitly analyze evaluation gaps and emerging needs in semantic, emotional, and alignment-based metrics, an area not addressed in these multimodal reviews.

## B  Analytical Framework Details

Below we provide full details of the corpus curation process, semantic taxonomy methodology, and experiment extraction pipeline.

### B.1  Semantic Taxonomy

To systematically capture the structure of the talking-head generation literature, we construct a hierarchical semantic taxonomy that organizes research papers according to thematic similarity.

The process begins by representing each paper's title and abstract as a numerical feature vector. We use a TF-IDF vectorizer with English stop-word removal and a vocabulary capped at 5,000 terms, producing a sparse document-term matrix of size $117 \times 5000$. This representation emphasizes terms that are particularly distinctive to each document relative to the full corpus, providing a discriminative foundation for uncovering latent thematic groupings.

Next, we apply hierarchical agglomerative clustering to the TF-IDF embeddings using Ward's linkage criterion. Ward's method minimizes intra-cluster variance when merging clusters, which tends to produce coherent and interpretable topic structures. A distance threshold is used to automatically determine the number of clusters, balancing granularity and interpretability.

To visualize the hierarchical organization, we generate a dendrogram using the `scipy.cluster.hierarchy.dendrogram` function. The leaf ordering preserves hierarchical relationships, placing semantically similar papers adjacent to one another. Beyond the raw clustering, we refine cluster identities using a large language model, which assigns human-readable thematic labels to each group, bridging data-driven organization with intuitive interpretability.

The resulting taxonomy provides a hierarchical map of research themes, showing both broad directions and fine-grained subtopics within the talking-head generation literature. By combining quantitative text analysis with expert-guided labeling, this approach enables insights into how methods, objectives, and modalities are distributed across the field, and highlights natural groupings that inform subsequent analyses of datasets, evaluation metrics, and emerging trends.

To provide a meaningful overview of the talking-head generation field, it is crucial to base our analysis on a corpus that is both comprehensive and representative. Talking-head research spans diverse subareas, including 2D and 3D face animation, audio-visual learning, portrait synthesis, and emerging generative paradigms such as NeRF- and diffusion-based models. Standard keyword searches or ad hoc selection risk overlooking influential contributions, particularly as high-impact work increasingly appears first as preprints. To address these challenges, we developed a multi-stage curation process that balances breadth, relevance, and community validation.

The process begins with broad retrieval, casting a wide net across the literature using domain-specific keywords (Talking Face, Talking Head, Visual Dubbing, Face Genertation, Lip Sync, Talker, Portrait, Talking Video, Head Synthesis, Face Reenactment, Wav2Lip, Talking Avatar, Lip Generation, Lip-Synchronization, Portrait Animation, Facial Animation, Lip Expert). Candidate papers are automatically collected from arXiv and enriched with structured metadata, including titles, authors, abstracts, submission history, and identifiers. Where available, we link papers to their code implementations on GitHub or Papers With Code, capturing not only conventional approaches but also newer, rapidly evolving paradigms. This initial stage ensures that diverse modalities, model families, and generative mechanisms are included, minimizing selection bias while capturing the full spectrum of methodological innovation.

To improve the reliability of the corpus, we distinguish peer-reviewed publications from arXiv-only preprints, verifying acceptance through Semantic Scholar and CrossRef. Peer-reviewed papers are treated as academically validated, while preprints are further filtered based on community impact, with only those accompanied by widely adopted implementations retained. Venue quality is also considered, using established top-tier lists in computer vision, machine learning, graphics, and multimedia to provide an objective proxy for academic rigor and community recognition. This approach allows the corpus to reflect both formal scholarly validation and practical influence within the research community.

Finally, to ensure thematic precision, all remaining papers undergo a manual semantic review. This step removes works outside the scope of talking-head generation, such as general video synthesis, speech-only models, or unrelated multimodal systems, ensuring that the final set of methods directly addresses the generation of realistic, expressive human head motion.

Through this carefully designed pipeline, the resulting corpus captures the major model families, diverse control modalities, and influential contributions that collectively define the current landscape of talking-head research. By integrating automated retrieval, community validation, venue assessment, and expert semantic review, the selection process establishes a high-quality foundation for subsequent analysis, including the construction of a semantic taxonomy, longitudinal study of datasets and metrics, and identification of emerging research trends. Table 1 shows the overview of selected key talking-head generation methods.

**Table 1: Overview of 117 key talking-head generation methods across clusters. GitHub star counts were collected on December 8, 2025. Refer to Figure 1 for cluster details.**

| Cluster | ArXiv ID | Method | Venue | Github | Stars | New Metrics |
|---|---|---|---|---|---|---|
| 1 | 2409.09292 | StyleTalk++ [204] | TPAMI2024 | No | No | No |
| 1 | 2403.06365 | Style2Talker [190] | AAAI2024 | No | No | No |
| 1 | 2310.07236 | AdaMesh [82] | TMM2025 | No | No | No |
| 1 | 2312.10877 | Mimic [102] | AAAI2024 | No | No | LDD |
| 1 | 2310.00434 | DiffPoseTalk [188] | TOG2024 | No | No | MOD |
| 1 | 2308.04830 | VAST [83] | ICCV2023 | No | No | No |
| 1 | 2301.01081 | StyleTalk [158] | AAAI2023 | Code | 520 | No |
| 1 | 2301.00023 | Imitator [194] | ICCV2023 | No | No | No |
| 2 | 2504.18087 | DICE-Talk [191] | ACM MM2025 | No | No | No |
| 2 | 2503.00495 | TexTalker [143] | CVPR2025 | No | No | No |
| 2 | 2403.06375 | FlowVQTalker [189] | CVPR2024 | No | No | No |
| 2 | 2309.04946 | EAT [13] | ICCV2023 | Code | 294 | No |
| 2 | 2306.08990 | EMOTE [93] | SIGGRAPH2023 | No | No | No |
| 2 | 2303.11089 | EmoTalk [173] | ICCV2023 | Code | 400 | EVE |
| 2 | 2305.02572 | HiEmoTalk [215] | CVPR2023 | No | No | No |
| 2 | 2205.01155 | EC-Talk [42] | IJCAI2022 | No | No | No |
| 2 | 2008.03592 | SpeechEmotion [99] | TMM2021 | Code | 172 | No |
| 2 | 2104.07452 | EVP [124] | CVPR2021 | No | No | No |

| Cluster | ArXiv ID | Method | Venue | Github | Stars | New Metrics |
|---|---|---|---|---|---|---|
| 3 | 2505.23290 | Wav2Sem [137] | CVPR2025 | No | No | No |
| 3 | 2412.09892 | VQTalker [152] | AAAI2025 | No | No | No |
| 3 | 2409.19143 | DCQL [107] | TVCG2025 | No | No | FDD; UPD; LPD; MPD |
| 3 | 2408.06010 | DEEPTalk [132] | AAAI2025 | No | No | FFD; Emo-FID |
| 3 | 2405.03121 | AniTalker [151] | ACM MM2024 | Code | 1600 | No |
| 3 | 2404.12888 | Learn2Talk [247] | TVCG2024 | No | No | SyncNet3D |
| 3 | 2312.02781 | PMMTalk [113] | TMM2024 | No | No | No |
| 3 | 2309.11306 | FaceDiffuser [187] | SIGGRAPH2023 | Code | 174 | Emo Diversity metric |
| 3 | 2309.04814 | Speech2Lip [210] | ICCV2023 | Code | 74 | No |
| 3 | 2306.10799 | SelfTalk [172] | ACM MM2023 | Code | 141 | LRP |
| 3 | 2304.08945 | DIRFA [209] | PR2023 | No | No | No |
| 3 | 2304.03199 | FADM [229] | CVPR2023 | Code | 75 | No |
| 3 | 2301.02379 | CodeTalker [53] | CVPR2023 | Code | 598 | FDD |
| 3 | 2303.17480 | SeeWhatUSay [200] | CVPR2023 | Code | 425 | WER |
| 3 | 2212.04248 | A2V-Diffusion [228] | ICCV2023 | No | No | SND |
| 3 | 2211.14758 | VideoReTalking [86] | SIGGRAPH2022 | No | No | No |
| 3 | 2104.08223 | MeshTalk [178] | ICCV2021 | Code | 394 | No |
| 3 | 2112.05329 | FaceFormer [12] | CVPR2022 | Code | 899 | No |
| 3 | 2108.07938 | FACIAL [231] | ICCV2021 | Code | 384 | No |
| 4 | 2507.06071 | MEDTalk [147] | ACM MM2025 | No | No | MLE; MEE; EIE; FRD |
| 4 | 2503.19383 | MVPortrait [146] | CVPR2025 | No | No | Variability |
| 4 | 2503.14517 | Cafe-Talk [81] | ICLR2025 | No | No | Control Rate (CR) |
| 4 | 2503.01715 | KeyFace [75] | CVPR2025 | No | No | LipScore; NSV_acc |
| 4 | 2501.01808 | MoEE [148] | CVPR2025 | No | No | No |
| 4 | 2410.23836 | Stereo-Talker [94] | TPAMI2025 | No | No | No |
| 4 | 2304.00334 | TalkCLIP [157] | TMM2025 | No | No | No |
| 4 | 2211.14506 | PDR-Talk [198] | CVPR2023 | Code | 97 | NLSE-C |
| 5 | 2412.01064 | FLOAT [131] | ICCV2025 | Code | 428 | No |
| 5 | 2503.18860 | HunyuanPortrait [217] | CVPR2025 | No | No | No |
| 5 | 2502.19894 | LCVD [110] | CVPR2025 | Code | 57 | No |
| 5 | 2412.00733 | Hallo3 [92] | Arxiv2025 | Code | 1340 | No |
| 5 | 2407.03168 | LivePortrait [109] | Arxiv2024 | Code | 17374 | No |
| 5 | 2401.08503 | Real3D-Portrait [222] | ICLR2024 | No | No | No |
| 5 | 2403.17694 | AniPortrait [208] | Arxiv2024 | Code | 5018 | No |
| 5 | 2311.15230 | GAIA [115] | ICLR2023 | No | No | No |
| 5 | 2308.12866 | ToonTalker [106] | ICCV2023 | No | No | No |
| 5 | 2307.10008 | MODA [153] | ICCV2023 | No | No | No |
| 5 | 2106.04185 | LipSync3D [134] | CVPR2021 | No | No | No |
| 6 | 2504.05746 | TAVCE [216] | TMM2025 | No | No | No |
| 6 | 2410.13726 | DAWN [85] | ICLR2024 | No | No | Degradation Rate |
| 6 | 2503.18429 | Teller [240] | CVPR2025 | No | No | No |
| 6 | 2501.14646 | SyncAnimation [154] | IJCAI2025 | No | No | EAR; Diversity of head motion |
| 6 | 2412.16915 | FADA [242] | CVPR2025 | No | No | No |
| 6 | 2412.09262 | LatentSync [135] | Arxiv2024 | Code | 5177 | No |
| 6 | 2411.19509 | Ditto [22] | ACM MM2025 | Code | 590 | No |
| 6 | 2411.16331 | Sonic [123] | CVPR2025 | Code | 3103 | No |
| 6 | 2411.16726 | EmotiveTalk [199] | CVPR2025 | No | No | No |
| 6 | 2411.15436 | ConsistentAvatar [218] | ACM MM2024 | No | No | No |
| 6 | 2410.10122 | MuseTalk [235] | Arxiv2024 | Code | 4999 | No |
| 6 | 2408.09384 | FD2Talk [219] | ACM MM2024 | No | No | No |
| 6 | 2408.01826 | GLDiTalker [145] | IJCAI2024 | No | No | No |
| 6 | 2407.05712 | MobilePortrait [126] | CVPR2025 | No | No | No |
| 6 | 2403.01901 | FaceChain-ImagineID [54] | CVPR2024 | Code | 9491 | No |
| 6 | 2403.19144 | MoDiTalker [20] | AAAI2025 | Code | 176 | No |
| 6 | 2411.09209 | JoyVASA [79] | Arxiv2024 | Code | 844 | No |
| 7 | 2312.10921 | AE-NeRF [136] | AAAI2024 | No | No | No |

| Cluster | ArXiv ID | Method | Venue | Github | Stars | New Metrics |
|---|---|---|---|---|---|---|
| 7 | 2307.09323 | ER-NeRF [140] | ICCV2023 | Code | 1223 | No |
| 7 | 2304.05097 | HiDe-NeRF [23] | CVPR2023 | No | No | AVD |
| 7 | 2301.13430 | GeneFace [59] | ICLR2023 | No | No | No |
| 7 | 2211.12368 | RT-NeRF-Talk [193] | IJCV2025 | No | No | No |
| 7 | 2103.11078 | AD-NeRF [14] | ICCV2021 | Code | 1066 | No |
| 8 | 2506.14742 | SyncTalk++ [170] | TPAMI2025 | No | No | No |
| 8 | 2311.17590 | SyncTalk [171] | CVPR2024 | Code | 1591 | No |
| 9 | 2305.06225 | DaGAN++ [118] | TPAMI2023 | Code | 996 | No |
| 9 | 2203.06605 | DaGAN [120] | CVPR2022 | Code | 996 | No |
| 10 | 2505.18096 | DualTalk [169] | CVPR2025 | No | No | P-FD; rPCC |
| 10 | 2504.19165 | IM-Portrait [144] | CVPR2025 | No | No | No |
| 10 | 2504.00665 | MGGTalk [105] | CVPR2025 | No | No | No |
| 10 | 2502.20387 | InsTaG [139] | CVPR2025 | Code | 154 | No |
| 10 | 2412.08504 | PointTalk [212] | AAAI2025 | No | No | No |
| 10 | 2412.00719 | MS-Codec [239] | CVPR2025 | No | No | No |
| 10 | 2409.13180 | FreeAvatar [175] | SIGGRAPH2024 | No | No | No |
| 10 | 2407.08136 | EchoMimic [84] | AAAI2025 | No | No | No |
| 10 | 2404.14037 | GaussianTalker [225] | ACM MM2024 | No | No | No |
| 10 | 2404.15264 | TalkingGaussian [138] | ECCV2024 | No | No | No |
| 10 | 2212.05005 | MemFace [192] | TPAMI2024 | No | No | No |
| 10 | 2303.17550 | DAE-Talker [11] | ACM MM2023 | No | No | No |
| 10 | 2305.00521 | StyleLipSync [130] | ICCV2023 | No | No | No |
| 10 | 2312.07385 | GSmoothFace [232] | TVCG2025 | No | No | No |
| 10 | 2304.10168 | HF-Controllable [104] | CVPR2023 | No | No | No |
| 10 | 2304.03275 | FCTalker [122] | ACM MM2023 | No | No | No |
| 10 | 2307.09906 | Implicit [119] | ICCV2023 | Code | 254 | No |
| 10 | 2305.08293 | IPTF [243] | CVPR2023 | Code | 737 | No |
| 10 | 2305.05445 | StyleSync [108] | CVPR2023 | No | No | No |
| 10 | 2301.03786 | DiffTalk [40] | CVPR2023 | Code | 469 | No |
| 10 | 2212.08062 | MetaPortrait [62] | CVPR2023 | Code | 547 | No |
| 10 | 2211.12194 | SadTalker [63] | CVPR2023 | Code | 13395 | No |
| 10 | 2303.03988 | DINet [236] | Arxiv2023 | Code | 1097 | No |
| 10 | 2211.09809 | SPACE [112] | ICCV2023 | No | No | No |
| 10 | 2211.00924 | SyncTalkFace [168] | AAAI2022 | No | No | No |
| 10 | 2209.04252 | StyleTalker [72] | ACM MM2022 | No | No | No |
| 10 | 2208.02210 | Free-HeadGAN [96] | TPAMI2023 | No | No | No |
| 10 | 2203.04036 | StyleHEAT [223] | ECCV2022 | Code | 656 | No |
| 10 | 2201.05986 | DCK [221] | TMM2022 | No | No | No |
| 10 | 2112.02749 | AVCT [203] | AAAI2022 | Code | 359 | No |
| 10 | 2012.08261 | HeadGAN [97] | ICCV2021 | No | No | No |
| 10 | 2107.09293 | Audio2Head [202] | IJCAI2021 | Code | 353 | No |
| 10 | 2104.14557 | LSR [159] | ICCV2021 | No | No | No |
| 10 | 2104.11116 | PC-AVS [244] | CVPR2021 | Code | 958 | Pose LMD |

## B.2 Datasets and Evaluation Metrics

To analyze experimental practices in talking-head generation, we examine a curated set of 117 representative papers spanning 2021-2025. For each paper, we extract the experiment section from its LaTeX source obtained via the arXiv API and process it with a structured GPT-based prompt to identify the datasets used, their roles in training or testing, the evaluation metrics used, and any newly proposed datasets or metrics.

The following prompt is used for dataset-metric extraction:

```
Briefly answer the following questions based on the cleaned Experiment section of the paper:

1. What datasets does this paper use?
2. Which datasets are used for training, and which for testing?
3. What evaluation metrics are used?
```

```
    Keep the answer concise and factual.
```

The following prompt is used for identifying newly proposed evaluation metrics:

```
    Please read the following paper content and answer ONLY the three points below:
    1. Did the paper propose any new evaluation metrics? Respond only with "Yes" or "No".
    2. If the answer is "Yes", briefly describe what each new metric is used for.
    3. For each new metric, indicate which evaluation category it belongs to. Use categories such as (but
    not limited to):
    · Visual Quality Metrics
    · Audio-Visual Sync Metrics
    · Geometry & Landmark Accuracy
    · Expression Naturalness
    · Driving Alignment

    Keep the answer concise and factual.
```

Extracted information is then normalized to unify naming conventions, resolving ambiguities such as VoxCeleb-2 versus VoxCeleb2 or LMD versus Landmark Distance, enabling reliable cross-paper analysis and longitudinal trend detection.

Datasets are categorized into four semantic groups, face images, speech audio, audio-visual, and emotion datasets, while evaluation metrics are mapped to five commonly used dimensions: visual quality, audio-visual synchronization, geometry and landmark accuracy, expression naturalness, and driving alignment. Newly proposed metrics are categorized in the same way, allowing us to track innovation across the field over time.

## C  Dataset Details

Table 2 summarizes the major datasets used in talking-head generation. Figure 5 presents the dataset usage. This figure summarizes the usage frequency of commonly adopted datasets in face image generation papers from 2021 to 2025. We select a representative set of widely used datasets and count their occurrences across the surveyed literature. To ensure consistency, datasets with identical sources or highly similar variants (*e.g.*, extensions or reprocessed versions) are merged under unified names, and only explicitly reported datasets are included. The statistics of evaluation metrics are computed using the same normalization and counting strategy. Below we provide some dataset details.

### C.1  Image-only

**AffectNet** [162] is a large-scale in-the-wild facial expression dataset collected from the Internet. Released in 2017, it contains approximately one million images, of which around 440K are manually annotated with basic emotion categories. The dataset exhibits diverse resolutions and unconstrained visual conditions, making it suitable for real-world facial expression analysis.

**CelebA** [155] is an in-the-wild face image dataset introduced in 2015, containing 202,599 images of 10,177 identities. Images are cropped to a typical size of approximately 178×218. The dataset was curated from publicly available Internet sources and organized by CUHK-MMLAB, and is widely used for identity modeling and facial attribute learning.

**CelebA-HQ** [127] is a high-quality subset of CelebA, constructed through processing and filtering the original dataset. It provides 30,000 high-resolution images, each at 1024×1024 resolution, and is commonly used for high-fidelity face synthesis and generative modeling.

**FFHQ** [128] (Flickr-Faces-HQ) is an in-the-wild high-resolution face dataset released by NVIDIA. It contains 70,000 images at 1024×1024 resolution, sourced from Flickr and other Internet repositories. FFHQ provides rich diversity in age, ethnicity, lighting, and background variations, and has become a standard benchmark for GAN-based image generation.

### C.2  Audio-only

**LibriSpeech** [167] is a large-scale speech corpus derived from LibriVox audiobooks, recorded in relatively clean indoor environments. Released in 2015, it includes approximately 1,000 hours of English speech from around 1,000 speakers. It is widely used for speech recognition, audio processing, and multimodal learning.

### C.3  Audio-visual

**VoxCeleb1** [30] is a large-scale audio-visual speaker dataset automatically collected from public YouTube interviews in in-the-wild conditions. Released in 2017, it contains over 150,000 utterances from 1,251 speakers, providing diverse real-world variations in pose, illumination, background, and speech content. It is widely used for speaker verification, cross-modal matching, and audio-visual synchronization research.

**VoxCeleb2** [7] is a significantly expanded version of VoxCeleb, also sourced from unconstrained YouTube videos. Released in 2018, it comprises more than 1 million utterances from 6,112 speakers (over 2,000 hours of audio-visual data), offering substantially richer variation in identity, speaking style, acoustic conditions, and visual appearance. It has become a standard benchmark for large-scale speaker recognition and robust audio-visual learning.

**HDTF** [64] is a high-resolution in-the-wild talking-face dataset collected from YouTube videos. Released in 2021, it includes over 16 hours of 720p-1080p audio-visual recordings from more than 300 speakers. The dataset contains natural head movements and expressive facial motion, making it suitable for talking-head generation, lip synchronization, and head-motion modeling.

**GRID** [9] is a lab-controlled audio-visual speech corpus consisting of 34 speakers, each reading 1,000 fixed-structure sentences (34,000 utterances in total). The dataset provides clean, synchronized audio and frontal facial video, making it a widely used benchmark for lip-reading, visual speech recognition, and speech-driven facial animation.

**TCD-TIMIT** [114] is a controlled audio-visual speech dataset recorded in a studio environment. It contains 6,913 phonetically diverse utterances from 62 speakers, captured with high-quality audio and synchronized frontal videos. The dataset originates from the Sigmedia group at Trinity College Dublin and supports research in audio-visual speech recognition, lip-reading, and speech-to-video synchronization.

**LRS2** [1] is a large-scale audio-visual speech dataset derived from BBC television broadcasts. Released around 2017, it contains approximately 224 hours of video with over 144,000 utterances, covering natural, unconstrained speaking scenarios. The dataset includes diverse face tracks with noticeable lip and head movements, and is widely used for lip-reading and audio-visual speech recognition.

**LRS3** [2] is an audio-visual corpus built from TED and TEDx YouTube videos collected in the wild. Released in 2018, it contains roughly 433-440 hours of English speech videos spanning thousands of speakers. The dataset offers rich variations in speaking style, facial appearance, pose, and motion, making it a benchmark for audio-visual speech recognition and talking-head modeling.

**LRS3-TED** [2] is an extended large-scale audio-visual dataset constructed from TED and TEDx videos, also released in 2018. It includes over 400 hours of face-track video accompanied by aligned audio, subtitles, and word-level annotations. The dataset captures natural facial, lip, and head motion, and is widely used for lip-reading and end-to-end audio-visual speech learning.

**LRW** [88] is a large-scale in-the-wild English lip-reading dataset constructed from BBC broadcast footage. Released in 2017, it contains over 500,000 short clips (each about 1.16 seconds, 29 frames) spanning 500 target words spoken by hundreds of speakers. The dataset provides substantial visual variability in pose, illumination, and speaking style, and has become a standard benchmark for visual speech recognition, lip-reading, and audio-visual synchronization research.

**VFHQ** [211] is a high-fidelity in-the-wild face video dataset collected from online interview and talk-show footage. Released in 2022, it consists of more than 16,000 high-quality clips with frame resolutions ranging from approximately 700×700 to 1000×1000. The dataset preserves rich facial detail, expression variation, and natural head motion, making it suitable for talking-head generation, face video restoration, and super-resolution tasks.

**TalkingHead-1KH** [205] is a large-scale in-the-wild talking-head video dataset curated from publicly available YouTube videos under permissive licenses. Released in 2021, it contains roughly 500,000 video clips, including around 80,000 clips with resolutions exceeding 512×512. The dataset offers diverse facial dynamics, lighting conditions, and head-motion patterns, supporting research on talking-head generation, lip synchronization, and free-view synthesis.

**CelebV-HQ** [67] is a high-quality large-scale face video dataset sourced from online celebrity videos collected in the wild. Introduced in 2022, it includes 35,666 curated clips spanning 15,653 identities, with all videos at a minimum of 512×512 resolution. Each clip is annotated with 83 human-labeled facial/action/emotion attributes. The dataset exhibits rich head pose, expression, and appearance variation, and is widely used for face-video generation, editing, and attribute modeling.

**CelebV-Text** [226] is a large-scale text-video face dataset constructed from web-crawled celebrity and public-domain videos. Released in 2023, it contains approximately 70,000 face-centric video clips (around 279 hours in total), with each clip paired with 20 textual descriptions capturing appearance, illumination, motion, emotion, and dynamic changes. It is a benchmark for text-to-video generation, expression/motion control, and multimodal semantic modeling.

**VOCASET** [91] is a controlled audio-4D facial motion dataset recorded in a studio environment using high-fidelity 3D scanning. It contains 480 speech-driven 4D facial motion sequences from 12 speakers (6 male, 6 female), each captured at 60 fps with synchronized audio. The dataset provides precise 3D facial mesh sequences and is widely used for speech-driven 3D facial animation and speech-to-motion modeling.

## C.4 Emotion

**CREMA-D** [4] is a controlled audio-visual emotional speech dataset recorded in a laboratory setting. It features 91 actors of diverse age and ethnicity portraying six basic emotions across multiple intensities, resulting in 7,442 short clips. The dataset includes audio-only, visual-only, and audio-visual modalities, making it valuable for emotion recognition and multimodal affect analysis.

**MEAD** [50] is a controlled lab-recorded multi-view emotional audio-visual dataset containing 60 actors. Released in 2020, it provides videos across 8 basic emotions and 3 intensity levels, captured simultaneously from 7 calibrated camera viewpoints. MEAD offers high-quality facial appearance and expression variations, supporting research on emotion-controllable talking-head generation and facial expression modeling.

**RAVDESS** [25] is a studio-recorded multimodal emotion dataset containing speech and song performances. It includes 7,356 audio and video recordings from 24 professional actors (12 male, 12 female), providing high-quality 16-bit 48kHz audio and 720p video. The dataset is widely used in speech emotion recognition, audiovisual affect modeling, and multimodal emotion analysis.

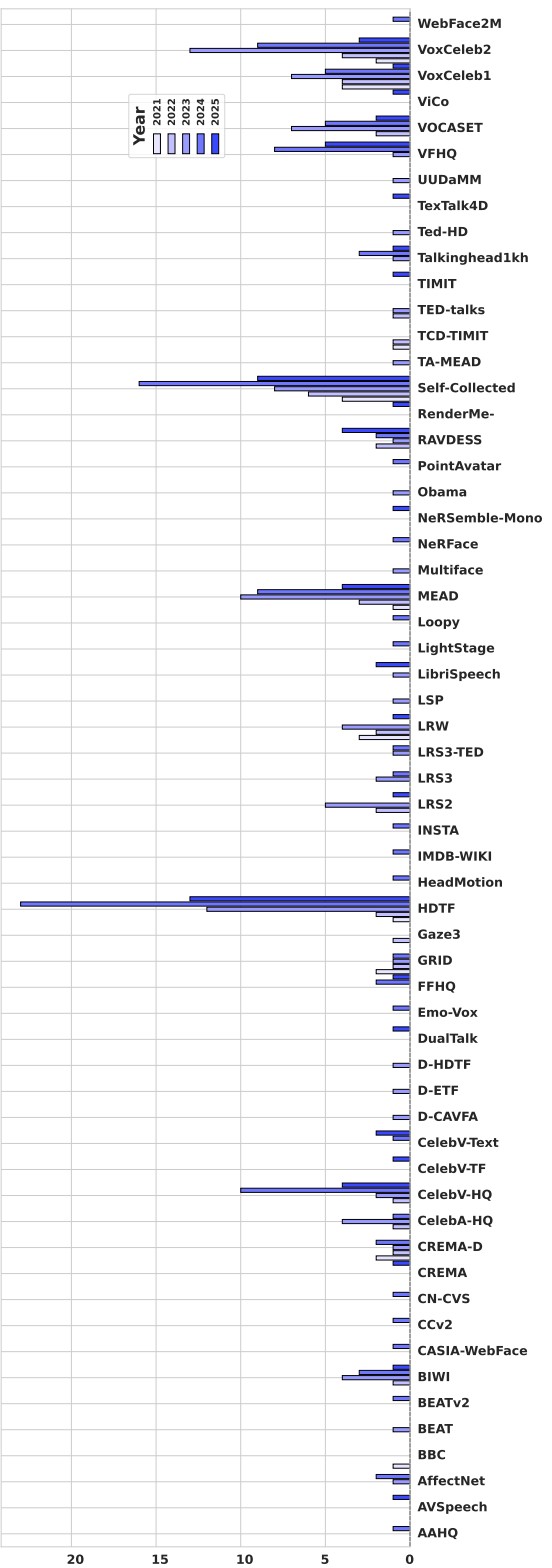

Figure 5: Dataset usage.

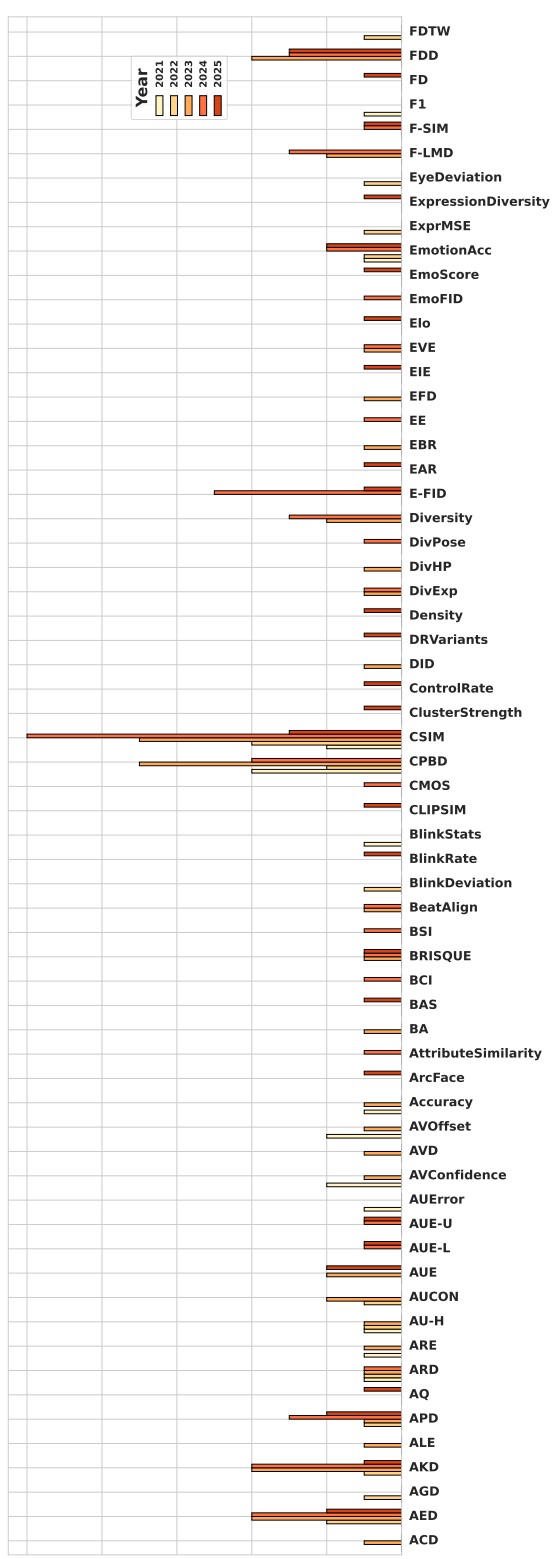

Figure 6: Metric usage (part 1)

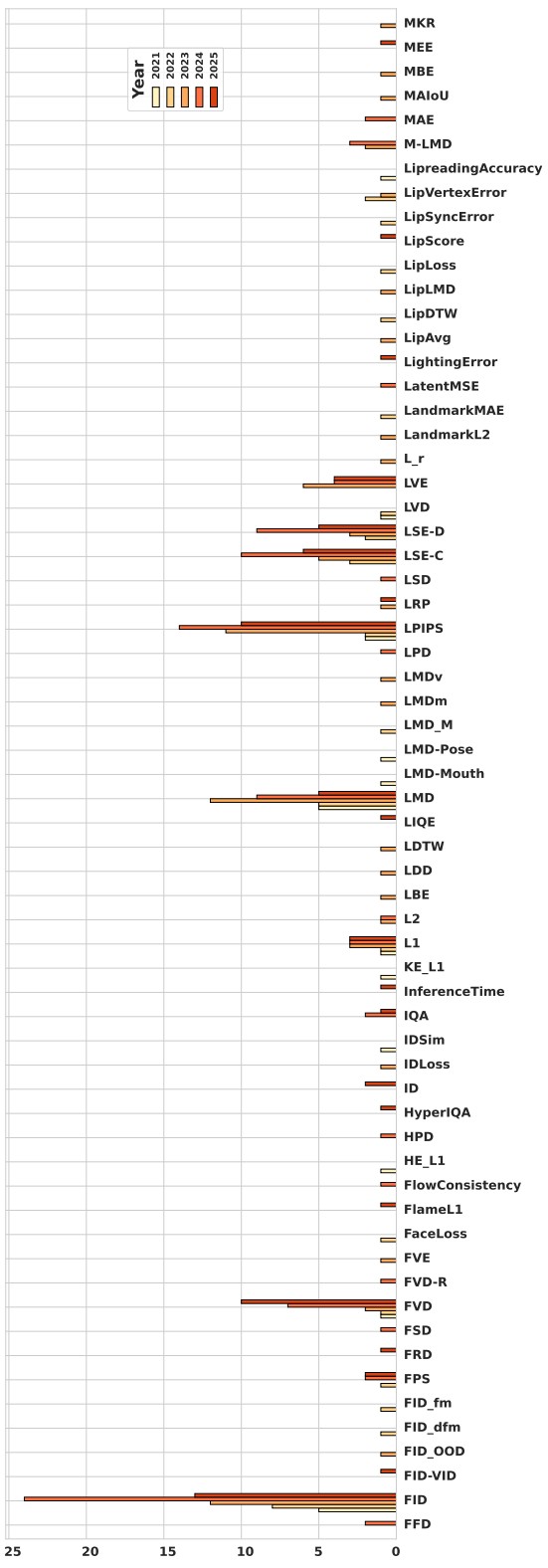

**Figure 7: Metric usage (part 2)**

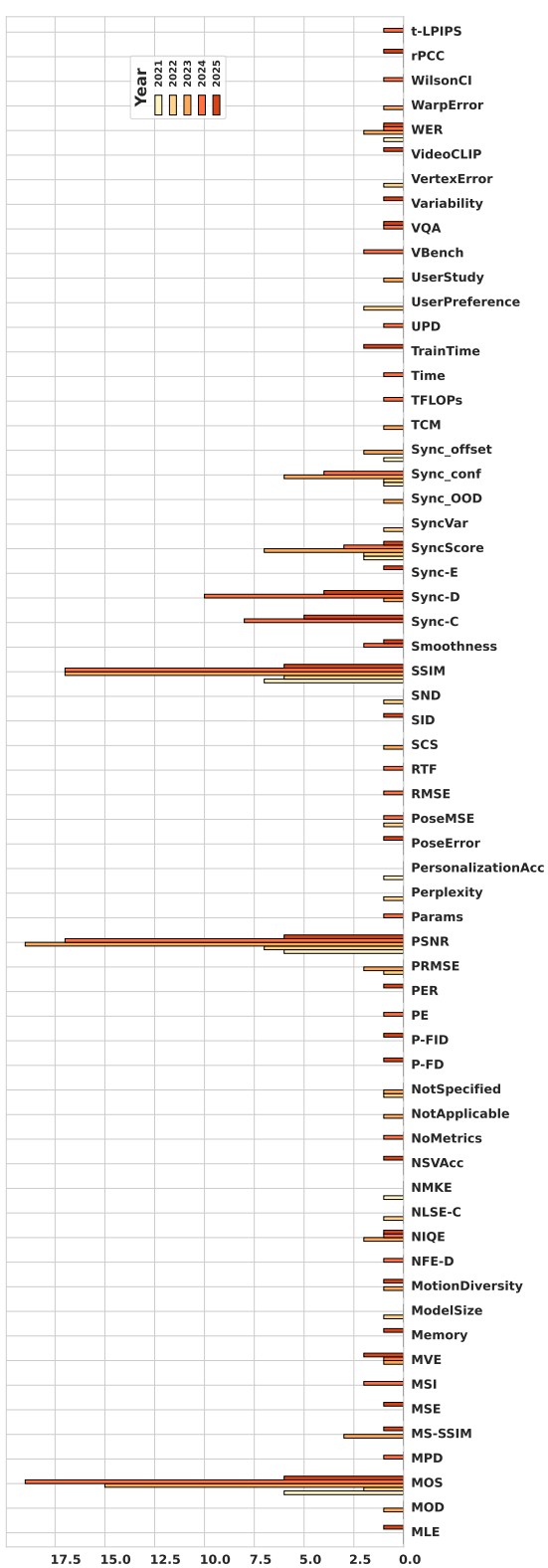

**Figure 8: Metric usage (part 3)**

| Dataset | Year | Speakers | Scale | Resolution | Source |
|---|---|---|---|---|---|
| AffectNet [162] | 2017 | N/A | ~1,000,000 images (440k labeled) | 425×425 | Internet in-the-wild |
| CelebA [155] | 2015 | 10,177 identities | 202,599 images | ~178×218 | Internet in-the-wild (CUHK-MMLAB) |
| CelebA-HQ [127] | 2017 | ~10,177 | 30,000 HQ images | 1024×1024 | Processed CelebA subset |
| FFHQ [128] | 2019 | ~70,000 | 70,000 images | 1024×1024 | Flickr & Internet |
| LibriSpeech [167] | 2015 | ~1,000 | 1,000 hours audio | N/A | LibriVox audiobooks |
| VoxCeleb1 [30] | 2017 | 1,251 | 150k utterances | 360×288, 720×576 | YouTube in-the-wild |
| VoxCeleb2 [7] | 2018 | 6,112 | 1M+ utterances (~2,000 h) | 224×224 | YouTube in-the-wild |
| HDTF [64] | 2021 | 300+ | 16 hours | 1280×720, 1920×1080 | YouTube in-the-wild |
| MEAD [50] | 2020 | 60 | Multi-emotion recorded sequences | 7-view HD cameras | Lab multi-view setup |
| GRID [9] | 2006 | 34 | 34,000 utterances | 360p-480p | Lab-controlled |
| TCD-TIMIT [114] | 2015 | 62 | 6,913 utterances | ~720p | Trinity College Dublin |
| LRS2 [1] | 2017 | N/A | 224 h, 144k utterances | 224×224 | BBC broadcasts |
| LRS3 [2] | 2018 | ~5,000 | 433-440 h | 224×224 | TED/TEDx videos |
| LRS3-TED [2] | 2018 | ~5,000 | 400+ h | 224×224 | TED/TEDx online videos |
| LRW [88] | 2017 | N/A | 500k+ clips (500 target words) | 224×224 | BBC broadcast |
| VFHQ [211] | 2022 | N/A | 16,000+ clips | 700×700-1000×1000 | Interviews / talk shows |
| TalkingHead-1KH [205] | 2021 | N/A | 500k video clips | ≥256p; 80k ≥512×512 | YouTube licensed |
| CelebV-HQ [67] | 2022 | 15,653 | 35,666 curated clips | ≥512×512 | Web celebrity videos |
| CelebV-Text [226] | 2023 | N/A | 70k clips, 20 text per clip | ≥512×512 | Web-crawled videos |
| VOCASET [91] | 2019 | 12 | 480 4D capture sequences | 3D mesh (60 fps) | Studio-controlled |
| CREMA-D [4] | 2014 | 91 | 7,442 clips | 480p-720p | Lab-controlled |
| RAVDESS [25] | 2018 | 24 | 7,356 recordings | 720p; 48kHz audio | Studio-recorded |

**Table 2: Summary of major datasets used in talking-head generation and audio-visual research.**

## D   Evaluation Metric Details

Figure 6, 7 and 8 present the metric usage. Below we provide some evaluation metric details.

### D.1   Visual Quality Metrics

**BRISQUE** [160] (Blind/Referenceless Image Spatial Quality Evaluator) is a no-reference IQA metric that assesses the level of distortion in an image without requiring a ground-truth reference. It models natural scene statistics (NSS) by applying locally normalized luminance transforms (MSCN) and extracting statistical features from luminance coefficients and pairwise products. These features are then fed into an SVR model trained on human-rated distorted images to predict a quality score. Lower BRISQUE values indicate images that are closer to natural, undistorted statistics, while higher values correspond to stronger perceptual degradation.

**CPBD** [165] (Cumulative Probability of Blur Detection) is a no-reference blur/sharpness metric that estimates the probability that blur in an image would be perceptually detectable by a human observer. It analyzes edge widths and edge contrast across the image, applies an HVS-based blur detection model, and aggregates the cumulative detection probability into a score in the range [0, 1]. Higher CPBD values indicate sharper images with less perceptible blur.

**NIQE** [161] (Natural Image Quality Evaluator) is a no-reference IQA metric that quantifies how much an image deviates from the statistical regularities of pristine natural images. The method learns a multivariate Gaussian model of NSS features extracted from high-quality images; for a test image, the same NSS features are computed and the Mahalanobis distance to the natural-image model is reported as the NIQE score. Lower NIQE indicates statistics closer to natural images, while higher values imply larger deviations and stronger distortions.

**MOS** [76] (Mean Opinion Score) is a subjective human evaluation protocol widely used for assessing perceptual quality in image, video, and audiovisual generation. Multiple human raters assign quality scores (typically on a 1-5 scale) to generated samples, and the final MOS is obtained by averaging scores across raters and/or samples. Higher MOS reflects better perceived realism and naturalness.

**SSIM** [51] (Structural Similarity Index) is a full-reference image quality metric used to measure the similarity between a generated image and its ground-truth counterpart in terms of structure, luminance, and contrast. It computes local statistics (mean, variance, covariance) within sliding windows to obtain luminance, contrast, and structural similarity components, which are then aggregated into a final SSIM score typically ranging from 0 to 1 (with 1 indicating perfect similarity). Higher SSIM values suggest fewer structural distortions. However, SSIM is primarily sensitive to low-level structural differences and does not capture semantic errors, identity inconsistencies, or temporal artifacts in video-based tasks such as talking-head generation.

**MS-SSIM** [207] (Multi-Scale Structural Similarity) is a perceptual image quality metric that extends SSIM by evaluating luminance, contrast, and structural similarity across multiple spatial scales. It is computed by iteratively applying low-pass filtering and downsampling to the input images, calculating SSIM at each scale, and combining the contrast-structure terms multiplicatively with a luminance term at the coarsest level. Scores range from 0 to 1, with higher values indicating greater perceptual similarity. However, MS-SSIM inherits limitations from SSIM: it is primarily designed for aligned pairs of natural images, may be insensitive to subtle spatial misalignments or temporal inconsistencies, and does not explicitly account for identity preservation, motion realism, or semantic correctness in talking-head generation.

**PSNR** [16] (Peak Signal-to-Noise Ratio) is a distortion-based full-reference metric that quantifies pixel-level fidelity between a generated image and its reference. It is derived from the mean squared error (MSE) as

$$\text{MSE} = \frac{1}{mn} \sum_{i=0}^{m-1} \sum_{j=0}^{n-1} (I(i,j) - K(i,j))^2, \tag{1}$$

$$\text{PSNR} = 10 \log_{10} \left( \frac{\text{MAX}^2}{\text{MSE}} \right), \tag{2}$$

where MAX denotes the maximum pixel value (*e.g.*, 255 for 8-bit images). Higher PSNR indicates closer pixel-wise correspondence. Despite its simplicity and widespread use, PSNR often correlates poorly with human perceptual judgments, especially in generative tasks where semantic errors, structural distortions, or texture inconsistencies may be visually significant but yield low MSE. Moreover, it assumes strict spatial alignment with ground truth and therefore provides limited insight into perceptual realism, identity fidelity, or temporal coherence in talking-head video synthesis.

**LPIPS** [43] (Learned Perceptual Image Patch Similarity) is a perceptual similarity metric that measures differences between images in the deep feature space of a pretrained CNN (*e.g.*, VGG or AlexNet). Images are passed through the network to extract multi-layer feature maps, which are channel-normalized and compared using L2 distances (optionally with learned channel-wise weights). Lower LPIPS scores indicate higher perceptual similarity. Compared with pixel-wise metrics such as PSNR or SSIM, LPIPS correlates better with human perception by being more sensitive to texture, semantics, and style differences. Nonetheless, its behavior depends on the backbone network and training domain and may not generalize perfectly to out-of-domain data such as synthetic or stylized talking-head frames.

**FID** [15] (Fréchet Inception Distance) is a widely used distribution-level metric for evaluating generative models. Real and generated images are fed into a pretrained Inception network (with the classification head removed), and their feature embeddings are modeled as multivariate Gaussian distributions. The Fréchet distance between these distributions serves as the FID score. Lower FID indicates that the distribution of generated images more closely matches that of real images. FID jointly reflects sample quality and diversity, making it more informative than single-image metrics such as SSIM or PSNR. However, it is sensitive to preprocessing, sample size, and the choice of feature extractor, and it does not account for temporal coherence or identity preservation in video generation tasks.

**E-FID** [227] extends FID by replacing the generic Inception feature extractor with a face- or expression-specific network trained on facial recognition or expression classification datasets. Distributional differences are then computed in this expression-aware embedding space. Lower E-FID scores indicate that the expression and facial attribute distributions of generated faces more closely match those of real faces. Like FID, E-FID is influenced by the choice and domain of the feature extractor and primarily captures frame-level appearance rather than temporal consistency.

**FVD** [197] (Fréchet Video Distance) is a distribution-level metric for assessing video generation quality. Real and generated videos are fed into a pretrained spatio-temporal recognition model (typically I3D) to extract embeddings that reflect both visual quality and temporal dynamics. These embeddings are modeled as multivariate Gaussians, and the Fréchet distance between them is used as the FVD score. Lower FVD values indicate that generated videos better match real ones in terms of appearance and motion. FVD captures temporal coherence beyond frame-based metrics such as FID, although it remains sensitive to the feature extractor's training domain and may not directly measure identity consistency, lip-sync accuracy, or fine-grained perceptual artifacts.

**MVE** [80] (Mean Vertex Error) is a standard geometric accuracy metric used in 3D face and head reconstruction, animation, and mesh-based talking-head generation. It computes the average Euclidean distance between corresponding vertices of the predicted mesh and the ground-truth mesh, typically after applying a rigid alignment step. MVE measures the fidelity of reconstructed geometry, including shape, pose, and expression. Lower MVE values indicate closer geometric correspondence to the ground truth, while higher values reflect larger deviations. MVE focuses solely on geometric structure and does not capture appearance realism, identity consistency, expression subtlety, or temporal smoothness.

**FPS** [166] (Frames Per Second) is a runtime performance metric that quantifies the speed of video generation or inference. It is defined as the number of frames produced per second:

$$\text{FPS} = \frac{\text{number of output frames}}{\text{wall-clock time (seconds)}}. \tag{3}$$

Higher FPS indicates faster generation and better suitability for real-time or high-throughput applications, whereas lower FPS reflects slower inference. FPS measures computational efficiency rather than visual quality; comparisons across methods must account for variations in hardware, resolution, batch size, and implementation optimizations.

## D.2 Audio-Visual Synchronization Metrics

**LSE-D / Sync-D** [8, 33] (Lip-Sync Error Distance) is a widely used audio-visual synchronization metric based on a pretrained lip-sync model such as SyncNet. For each audio-video segment, the mouth-region video frames and the corresponding audio snippet are fed into the sync network to obtain video and audio embeddings. The metric is computed as the average L2 distance between these embeddings across all temporal windows. Lower LSE-D values indicate closer alignment between lip movements and speech, while higher values reflect poorer synchronization. LSE-D provides an automatic and quantitative measure of lip-sync quality and is also used as a supervisory loss in some training pipelines. When the generated video quality is poor (low resolution/cropping/compression/blurring), embedding extraction may be unstable, making LSE-D unreliable. Furthermore, multiple studies [33, 176] have shown that LSE-D/LSE-C have limited correlation with human subjective evaluation (MOS).

**LSE-C / Sync-C** [8, 33] (Lip-Sync Error Confidence) is the confidence-based counterpart to LSE-D and also relies on a pretrained SyncNet-style model. Given an audio-video pair, the model outputs a synchronization confidence score (*e.g.*, cosine similarity, logits, or softmax-normalized probabilities) that reflects how likely the audio and lip motions are aligned. The final LSE-C score is obtained by averaging (or aggregating via median/percentile) confidence scores across time. Higher LSE-C values indicate stronger predicted alignment between speech and lip motion. Similar to LSE-D, LSE-C offers an automatic quantitative indicator of synchronization quality but does not assess appearance quality, expression naturalness, head motion, or identity stability, and its correlation with human subjective judgments is known to be limited.

**WER** [77, 179] (Word Error Rate) is a standard metric for evaluating the correctness of recognized or generated speech content in speech-driven talking-head systems. It measures the edit distance between the system output (hypothesis) and the ground-truth transcript, defined as:

$$\text{WER} = \frac{S + D + I}{N},$$

where $S$, $D$, and $I$ denote the number of substitutions, deletions, and insertions, respectively, and $N$ is the total number of words in the reference. Lower WER indicates more accurate speech content reproduction.

## D.3 Geometry & Landmark Accuracy

**L1 (Mean Absolute Error, MAE)** [117] measures the pixel-wise absolute difference between a generated image or video frame and its ground-truth counterpart. It is computed by taking the absolute value of the difference at each corresponding pixel, then averaging across all pixels (and optionally color channels or frames). Lower L1 values indicate closer pixel-level agreement with the reference.

**LMD (Landmark Distance)** [33] is a geometric accuracy metric used to assess how well the synthesized facial or lip motion matches real motion. Facial landmarks are first detected on both the generated and ground-truth frames, and the Euclidean distance between each pair of corresponding landmarks is computed (optionally normalized by face size or inter-ocular distance). The final score is obtained by averaging distances over all points and frames. Lower LMD values indicate better landmark alignment.

**F-LMD (Fréchet Landmark Distance)** extends landmark-based evaluation to the distribution level. Landmark features extracted from generated and real videos are modeled as multivariate Gaussian distributions, and the Fréchet distance between these distributions is computed, analogous to FID in image synthesis. Lower F-LMD values indicate closer alignment between the geometric motion distributions of generated and real faces.

**M-LMD (Mean Landmark Distance)** refers to the average Euclidean distance between corresponding facial landmarks across frames, often normalized by facial scale (*e.g.*, inter-ocular distance). Despite naming differences across papers, M-LMD is mathematically equivalent to landmark-based mean distance measures such as LMD, and it evaluates how closely generated facial or lip motion follows the ground truth.

**AVConfidence (Audio-Visual Confidence)** measures the alignment between the generated video (typically the lip or mouth region) and its corresponding audio using a pretrained audio-visual synchronization network such as SyncNet. The audio and video segments are encoded into embeddings, and their similarity (*e.g.*, cosine similarity or another confidence score) is computed over sliding windows or the full clip. The final AVConfidence is obtained by averaging (or taking the maximum of) these similarity scores. Higher values indicate stronger audio-lip synchrony. This metric is commonly used both for assessing lip-sync quality and for filtering misaligned samples during dataset preprocessing.

**AVOffset (Audio-Visual Offset)** quantifies the temporal misalignment between the speech signal and the corresponding lip or mouth motion. Using an AV-sync model (*e.g.*, SyncNet), the audio embedding is compared against video embeddings at multiple temporal shifts. For each time offset, a similarity or confidence score is computed, and the offset that yields the highest similarity (or lowest distance) is taken as the estimated time lag. The absolute offset, averaged across windows or frames, is reported as AVOffset. Values closer to zero indicate more accurate temporal synchronization. This metric is widely used to detect and correct audio-video drift in in-the-wild videos during preprocessing.

## D.4 Expression Naturalness

**EmotionAcc** [42, 54] is an accuracy-based metric that evaluates how well the generated face preserves or expresses target emotional states in talking-head generation or facial reenactment. It is typically computed by applying a pretrained emotion recognition classifier (*e.g.*, trained

to predict discrete categories such as happy, sad, angry, etc.) to both the generated frames and the reference (or target-labeled) frames, and reporting the proportion of frames where the predicted emotion matches the ground-truth or target emotion. Some methods additionally employ human raters to assess perceived emotional correctness.

**Diversity** [29] measures the variability of generated outputs across different samples, typically in terms of appearance attributes such as identity, pose, and expression, rather than realism alone. It is commonly computed as the average pairwise distance between feature embeddings extracted from a pretrained network (*e.g.*, a face recognition or image encoder). Higher average distances indicate more diverse outputs, while lower values suggest redundancy or mode collapse.

**Smoothness** [24] evaluates whether the motion or appearance of consecutive frames in a generated video changes smoothly over time, without jitter, flicker, or abrupt transitions. A typical formulation computes first- or second-order temporal differences in pixel space, feature space, or geometric representations such as landmarks, mesh vertices, or optical flow (*e.g.*, $\|x_t - x_{t-1}\|$ or $\|x_t - 2x_{t-1} + x_{t-2}\|$). These values are then averaged across frames to produce a Smoothness score. Lower values indicate smoother and more temporally consistent motion. Recent work also proposes more advanced temporal metrics, such as Fréchet Video Motion Distance (FVMD), which compares distributions of motion trajectories between real and generated videos, extending temporal evaluation beyond simple frame differencing.

**PRMSE (Pose RMSE)** [73] measures the accuracy of generated or predicted head pose relative to the ground truth in talking-head synthesis or facial animation. For each frame, yaw, pitch, and roll (and optionally translation parameters) are estimated for both the generated and reference frames using a consistent pose estimator. The squared differences between corresponding pose parameters are averaged across frames and parameters, and the square root of this mean yields the PRMSE. Lower PRMSE values indicate more accurate head-pose reproduction.

## D.5   Driving Alignment

**ARD (Absolute Relative Difference)** measures the relative error between predicted values (*e.g.*, depth, scale, motion magnitude, or 3D reconstruction parameters) and ground-truth values. For each sample $x$ (ground truth) and its prediction $\hat{x}$, ARD computes $|\hat{x} - x|/|x|$ and averages this ratio across all samples. Lower ARD values indicate closer agreement between prediction and ground truth in a relative sense.

**AKD (Average Keypoint Distance)** evaluates geometric accuracy by comparing facial or head keypoints extracted from generated frames and corresponding ground-truth frames using the same landmark detector. The Euclidean distances between corresponding 2D or 3D keypoints are computed and averaged across all points and frames. Lower AKD implies better alignment of facial motion, pose, and expression.

**APD (Average Pose Distance)** [61] quantifies the discrepancy between pose parameters estimated from generated frames and from the driving or reference frames. Pose may include head rotation (yaw, pitch, roll), translation, or 3DMM-based pose parameters. The distance between pose vectors is computed per frame and averaged across the sequence. Lower APD indicates more accurate reproduction of head pose and motion.

**AED (Average Expression Distance)** [18] measures how closely the generated facial expression matches the expression of the driving or target frame. Expression representations, typically 3DMM expression coefficients, dense landmarks, or other expression parameters, are extracted for both generated and reference frames, and their distances are averaged across frames. Lower AED indicates better alignment of facial expression and mouth motion dynamics.

**FDD (Face / Motion Dynamics Deviation)** [22] assesses temporal motion consistency by measuring how closely the dynamic trajectories of facial geometry (*e.g.*, 3D mesh vertices or facial keypoints) in generated videos follow those in the ground-truth sequence. For each frame, geometric differences between predicted and real representations are computed (*e.g.*, via L2 distance), and aggregated across time. Lower FDD reflects more faithful and natural facial-motion dynamics.

## E   Future Directions and Limitations

## E.1   Limitations of Text-Based Clustering and LLM-Based Labeling

The resulting clustering is inherently influenced by the vocabulary choices adopted in paper titles and abstracts, as TF-IDF representations capture lexical co-occurrence patterns rather than underlying methodological equivalence. Variations in terminology across authors, research communities, and publication periods may therefore affect fine-grained cluster boundaries. In addition, while large language models are used solely for post-hoc cluster labeling rather than structure generation, their inherent priors can influence the phrasing and emphasis of semantic summaries. Importantly, these factors primarily impact interpretability at the label or boundary level, while the overall hierarchical organization remains largely consistent with the manually derived taxonomy.

## E.2   Limitations Due to Public Data Availability

This survey is necessarily restricted to publicly available research and may therefore omit proprietary industrial systems or unpublished benchmarks. While such industrial practices can play an important role in advancing the field, their reliance on private datasets, undisclosed architectures, or non-reproducible evaluation protocols limits their suitability for systematic analysis. To ensure transparency, comparability, and longitudinal consistency, we deliberately focus on reproducible and publicly accessible research practices. As a result, the proposed taxonomy and trend analysis are intended to reflect the evolution of open and verifiable academic research, rather than to provide a comprehensive account of closed industrial developments.

### E.3 Reporting Inconsistencies and Extraction Limitations

The extraction process in this survey is inherently influenced by how original papers report their experimental setups and evaluation results, reflecting a broader lack of standardization in experimental reporting within the talking-head generation community. In practice, identical evaluation metrics are often described using different terminology, and experimental settings are reported with varying levels of granularity, which increases the difficulty of systematic information extraction and normalization and may lead to occasional omissions. To mitigate these effects, we adopt a combination of unified mapping rules and manual verification to ensure a reasonably robust and consistent extraction strategy across studies. Nevertheless, we acknowledge that fully addressing this issue requires community-level efforts beyond the scope of this work. Accordingly, we emphasize in the future outlook the potential value of establishing standardized experimental reporting templates to improve the completeness, comparability, and reproducibility of large-scale survey analyses and benchmarking efforts.

### E.4 Emerging Ethical Standards and Misuse Prevention

Although this work discusses issues related to deepfake technologies and demographic bias, concrete and widely accepted standards for misuse prevention and ethical risk assessment remain under active development. Current discussions surrounding deepfake misuse, bias amplification, and content provenance are still exploratory in nature, and no unified evaluation framework has yet been established. Accordingly, the goal of this survey is not to propose a mature ethical assessment methodology, but rather to explicitly identify misuse prevention as an important and currently under-standardized research frontier. By systematically reviewing existing debates and emerging perspectives, we aim to raise awareness within the research community of the significance and complexity of these ethical challenges, and to encourage further collective efforts toward developing robust and actionable ethical standards.

### E.5 Demographic Granularity and Benchmark Coverage

Although recent studies increasingly acknowledge demographic imbalance as a critical factor affecting model performance, the demographic composition of existing talking-head benchmarks remains underreported. As a result, it is difficult to assess whether current evaluation practices systematically under-represent certain populations across gender, age, or ethnicity. This limitation also affects our longitudinal analysis of 117 representative papers, as most datasets do not disclose sufficiently detailed demographic metadata. Future benchmark construction and reporting efforts would benefit from standardized demographic annotations, which would enable more rigorous analysis of fairness, bias amplification, and cross-demographic generalization in talking-head generation systems

### E.6 Human Evaluation Metadata and Subjective Reliability

Human evaluation remains an essential component for assessing perceptual quality, expressiveness, and naturalness in talking-head generation. While several studies recommend Mean Opinion Score (MOS) evaluations with at least 30 raters, detailed metadata such as inter-rater reliability, rater expertise, and annotation consistency are rarely reported in a standardized manner. This lack of transparency limits large-scale meta-analysis and complicates the interpretation of subjective scores. Moreover, recent evidence suggests that correlations between traditional MOS and emerging semantic metrics, such as lip-sync error scores (e.g., LSE-C/D), are often weak. Future work should place greater emphasis on reporting detailed human evaluation protocols and reliability statistics, facilitating deeper investigation into how subjective judgments align with objective and semantic evaluation metrics.

### E.7 Toward Unified Head-Body Dynamics and Semantic Grounding

Most existing talking-head generation methods focus exclusively on facial motion, often neglecting the broader context of body dynamics, gaze behavior, and emotional expression. However, recent trends suggest a growing interest in unified generative frameworks that integrate head motion with upper-body gestures and posture. In parallel, there is increasing recognition of the need for richer semantic grounding, ensuring that generated motions are not only synchronized with audio but also contextually appropriate for the spoken content. Future research should explore more rigorous linguistic and semantic evaluation metrics that assess whether generated behaviors align with dialogue intent, discourse context, and affective cues, rather than relying solely on low-level synchronization measures.

### References for Appendix

[69] Fakhar Abbas and Araz Taeihagh. 2024. Unmasking deepfakes: A systematic review of deepfake detection and generation techniques using artificial intelligence. *Expert Systems with Applications* 252 (2024), 124260.

[1] Triantafyllos Afouras, Joon Son Chung, Andrew Senior, Oriol Vinyals, and Andrew Zisserman. 2018. Deep audio-visual speech recognition. In *IEEE Conference on Computer Vision and Pattern Recognition (CVPR)*. 7099–7108.

[2] Triantafyllos Afouras, Joon Son Chung, and Andrew Zisserman. 2018. LRS3-TED: a large-scale dataset for visual speech recognition. *arXiv preprint arXiv:1809.00496* (2018).

[72] Mohammed M Alghamdi, He Wang, Andrew J Bulpitt, and David C Hogg. 2022. Talking head from speech audio using a pre-trained image generator. In *Proceedings of the 30th ACM International Conference on Multimedia*. 5228–5236.

[73] Andrea Asperti and Daniele Filippini. 2023. Deep learning for head pose estimation: A survey. *SN Computer Science* 4, 4 (2023), 349.

[74] Tadas Baltrušaitis, Chaitanya Ahuja, and Louis-Philippe Morency. 2018. Multimodal machine learning: A survey and taxonomy. *IEEE transactions on pattern analysis and machine intelligence* 41, 2 (2018), 423–443.

[75] Antoni Bigata, Michał Stypułkowski, Rodrigo Mira, Stella Bounareli, Konstantinos Vougioukas, Zoe Landgraf, Nikita Drobyshev, Maciej Zieba, Stavros Petridis, and Maja Pantic. 2025. Keyface: Expressive audio-driven facial animation for long sequences via keyframe interpolation. In *Proceedings of the Computer Vision and Pattern Recognition Conference*. 5477–5488.

[76] Sebastian Bosse, Dominique Maniry, Klaus-Robert Müller, Thomas Wiegand, and Wojciech Samek. 2017. Deep neural networks for no-reference and full-reference image quality assessment. *IEEE Transactions on image processing* 27, 1 (2017), 206–219.

[77] Herve A Bourlard and Nelson Morgan. 2012. *Connectionist speech recognition: a hybrid approach.* Vol. 247. Springer Science & Business Media.

[4] Houwei Cao, David G Cooper, Michael K Keutmann, Ruben C Gur, Ani Nenkova, and Ragini Verma. 2014. Crema-d: Crowd-sourced emotional multimodal actors dataset. *IEEE transactions on affective computing* 5, 4 (2014), 377–390.

[79] Xuyang Cao, Guoxin Wang, Sheng Shi, Jun Zhao, Yang Yao, Jintao Fei, and Minyu Gao. 2024. JoyVASA: portrait and animal image animation with diffusion-based audio-driven facial dynamics and head motion generation. *arXiv preprint arXiv:2411.09209* (2024).

[80] Zenghao Chai, Haoxian Zhang, Jing Ren, Di Kang, Zhengzhuo Xu, Xuefei Zhe, Chun Yuan, and Linchao Bao. 2022. Realy: Rethinking the evaluation of 3d face reconstruction. In *European conference on computer vision.* Springer, 74–92.

[81] Hejia Chen, Haoxian Zhang, Shoulong Zhang, Xiaoqiang Liu, Sisi Zhuang, Yuan Zhang, Pengfei Wan, Di Zhang, and Shuai Li. 2025. Cafe-talk: Generating 3d talking face animation with multimodal coarse-and fine-grained control. *arXiv preprint arXiv:2503.14517* (2025).

[82] Liyang Chen, Weihong Bao, Shun Lei, Boshi Tang, Zhiyong Wu, Shiyin Kang, Haozhi Huang, and Helen Meng. 2025. AdaMesh: Personalized Facial Expressions and Head Poses for Adaptive Speech-Driven 3D Facial Animation. *IEEE Transactions on Multimedia* (2025).

[83] Liyang Chen, Zhiyong Wu, Runnan Li, Weihong Bao, Jun Ling, Xu Tan, and Sheng Zhao. 2023. VAST: vivify your talking avatar via zero-shot expressive facial style transfer. In *Proceedings of the IEEE/CVF International Conference on Computer Vision.* 2977–2987.

[84] Zhiyuan Chen, Jiajiong Cao, Zhiquan Chen, Yuming Li, and Chenguang Ma. 2025. Echomimic: Lifelike audio-driven portrait animations through editable landmark conditions. In *Proceedings of the AAAI Conference on Artificial Intelligence*, Vol. 39. 2403–2410.

[85] Hanbo Cheng, Limin Lin, Chenyu Liu, Pengcheng Xia, Pengfei Hu, Jiefeng Ma, Jun Du, and Jia Pan. 2024. DAWN: Dynamic Frame Avatar with Non-autoregressive Diffusion Framework for Talking Head Video Generation. *arXiv preprint arXiv:2410.13726* (2024).

[86] Kun Cheng, Xiaodong Cun, Yong Zhang, Menghan Xia, Fei Yin, Mingrui Zhu, Xuan Wang, Jue Wang, and Nannan Wang. 2022. Videoretalking: Audio-based lip synchronization for talking head video editing in the wild. In *SIGGRAPH Asia 2022 Conference Papers.* 1–9.

[7] Joon Son Chung, Arsha Nagrani, and Andrew Zisserman. 2018. Voxceleb2: Deep speaker recognition. *arXiv preprint arXiv:1806.05622* (2018).

[88] Joon Son Chung and Andrew Zisserman. 2017. Lip reading in the wild. In *Computer Vision–ACCV 2016: 13th Asian Conference on Computer Vision, Taipei, Taiwan, November 20-24, 2016, Revised Selected Papers, Part II 13.* Springer, 87–103.

[8] Joon Son Chung and Andrew Zisserman. 2017. Out of time: automated lip sync in the wild. In *Computer Vision–ACCV 2016 Workshops: ACCV 2016 International Workshops, Taipei, Taiwan, November 20-24, 2016, Revised Selected Papers, Part II 13.* Springer, 251–263.

[9] Martin Cooke, Jon Barker, Stuart Cunningham, and Xu Shao. 2006. An audio-visual corpus for speech perception and automatic speech recognition. *The Journal of the Acoustical Society of America* 120, 5 (2006), 2421–2424.

[91] Daniel Cudeiro, Timo Bolkart, Cassidy Laidlaw, Anurag Ranjan, and Michael J Black. 2019. Capture, learning, and synthesis of 3D speaking styles. In *Proceedings of the IEEE/CVF conference on computer vision and pattern recognition.* 10101–10111.

[92] Jiahao Cui, Hui Li, Yun Zhan, Hanlin Shang, Kaihui Cheng, Yuqi Ma, Shan Mu, Hang Zhou, Jingdong Wang, and Siyu Zhu. 2025. Hallo3: Highly dynamic and realistic portrait image animation with video diffusion transformer. In *Proceedings of the Computer Vision and Pattern Recognition Conference.* 21086–21095.

[93] Radek Daněček, Kiran Chhatre, Shashank Tripathi, Yandong Wen, Michael Black, and Timo Bolkart. 2023. Emotional speech-driven animation with content-emotion disentanglement. In *SIGGRAPH Asia 2023 Conference Papers.* 1–13.

[94] Xiang Deng, Youxin Pang, Xiaochen Zhao, Chao Xu, Lizhen Wang, Hongjiang Xiao, Shi Yan, Hongwen Zhang, and Yebin Liu. 2025. Stereo-talker: Audio-driven 3d human synthesis with prior-guided mixture-of-experts. *IEEE Transactions on Pattern Analysis and Machine Intelligence* (2025).

[10] Ramamurthy Dhanyalakshmi, Gabriel Stoian, Daniela Danciulescu, and Duraisamy Jude Hemanth. 2025. A Survey on Face-Swapping Methods for Identity Manipulation in Deepfake Applications. *IET Image Processing* 19, 1 (2025), e70132.

[96] Michail Christos Doukas, Evangelos Ververas, Viktoriia Sharmanska, and Stefanos Zafeiriou. 2023. Free-headgan: Neural talking head synthesis with explicit gaze control. *IEEE Transactions on Pattern Analysis and Machine Intelligence* 45, 8 (2023), 9743–9756.

[97] Michail Christos Doukas, Stefanos Zafeiriou, and Viktoriia Sharmanska. 2021. Headgan: One-shot neural head synthesis and editing. In *Proceedings of the IEEE/CVF International conference on Computer Vision.* 14398–14407.

[11] Chenpeng Du, Qi Chen, Tianyu He, Xu Tan, Xie Chen, Kai Yu, Sheng Zhao, and Jiang Bian. 2023. Dae-talker: High fidelity speech-driven talking face generation with diffusion autoencoder. In *Proceedings of the 31st ACM International Conference on Multimedia.* 4281–4289.

[99] Sefik Emre Eskimez, You Zhang, and Zhiyao Duan. 2021. Speech driven talking face generation from a single image and an emotion condition. *IEEE Transactions on Multimedia* 24 (2021), 3480–3490.

[12] Yingruo Fan, Zhaojiang Lin, Jun Saito, Wenping Wang, and Taku Komura. 2022. Faceformer: Speech-driven 3d facial animation with transformers. In *Proceedings of the IEEE/CVF conference on computer vision and pattern recognition.* 18770–18780.

[101] Tharindu Fernando, Darshana Priyasad, Sridha Sridharan, Arun Ross, and Clinton Fookes. 2025. Face Deepfakes–A Comprehensive Review. *arXiv preprint arXiv:2502.09812* (2025).

[102] Hui Fu, Zeqing Wang, Ke Gong, Keze Wang, Tianshui Chen, Haojie Li, Haifeng Zeng, and Wenxiong Kang. 2024. Mimic: Speaking style disentanglement for speech-driven 3d facial animation. In *Proceedings of the AAAI conference on artificial intelligence*, Vol. 38. 1770–1777.

[13] Yuan Gan, Zongxin Yang, Xihang Yue, Lingyun Sun, and Yi Yang. 2023. Efficient emotional adaptation for audio-driven talking-head generation. In *Proceedings of the IEEE/CVF International Conference on Computer Vision.* 22634–22645.

[104] Yue Gao, Yuan Zhou, Jinglu Wang, Xiao Li, Xiang Ming, and Yan Lu. 2023. High-fidelity and freely controllable talking head video generation. In *Proceedings of the IEEE/CVF conference on computer vision and pattern recognition.* 5609–5619.

[105] Shengjie Gong, Haojie Li, Jiapeng Tang, Dongming Hu, Shuangping Huang, Hao Chen, Tianshui Chen, and Zhuoman Liu. 2025. Monocular and Generalizable Gaussian Talking Head Animation. In *Proceedings of the Computer Vision and Pattern Recognition Conference.* 5523–5534.

[106] Yuan Gong, Yong Zhang, Xiaodong Cun, Fei Yin, Yanbo Fan, Xuan Wang, Baoyuan Wu, and Yujiu Yang. 2023. ToonTalker: Cross-domain face reenactment. In *Proceedings of the IEEE/CVF international conference on computer vision.* 7690–7700.

[107] Chunzhi Gu, Shigeru Kuriyama, and Katsuya Hotta. 2025. Diverse Code Query Learning for Speech-Driven Facial Animation. *IEEE Transactions on Visualization and Computer Graphics* (2025).

[108] Jiazhi Guan, Zhanwang Zhang, Hang Zhou, Tianshu Hu, Kaisiyuan Wang, Dongliang He, Haocheng Feng, Jingtuo Liu, Errui Ding, Ziwei Liu, et al. 2023. Stylesync: High-fidelity generalized and personalized lip sync in style-based generator. In *Proceedings of the IEEE/CVF Conference on Computer Vision and Pattern Recognition.* 1505–1515.

[109] Jianzhu Guo, Dingyun Zhang, Xiaoqiang Liu, Zhizhou Zhong, Yuan Zhang, Pengfei Wan, and Di Zhang. 2024. Liveportrait: Efficient portrait animation with stitching and retargeting control. *arXiv preprint arXiv:2407.03168* (2024).

[110] Mingtao Guo, Guanyu Xing, and Yanli Liu. 2025. High-Fidelity Relightable Monocular Portrait Animation with Lighting-Controllable Video Diffusion Model. In *Proceedings of the Computer Vision and Pattern Recognition Conference.* 228–238.

[14] Yudong Guo, Keyu Chen, Sen Liang, Yong-Jin Liu, Hujun Bao, and Juyong Zhang. 2021. Ad-nerf: Audio driven neural radiance fields for talking head synthesis. In *Proceedings of the IEEE/CVF international conference on computer vision.* 5784–5794.

[112] Siddharth Gururani, Arun Mallya, Ting-Chun Wang, Rafael Valle, and Ming-Yu Liu. 2023. Space: Speech-driven portrait animation with controllable expression. In *Proceedings of the ieee/cvf international conference on computer vision.* 20914–20923.

[113] Tianshun Han, Shengnan Gui, Yiqing Huang, Baihui Li, Lijian Liu, Benjia Zhou, Ning Jiang, Quan Lu, Ruicong Zhi, Yanyan Liang, et al. 2024. PMMTalk : Speech-Driven 3D Facial Animation from Complementary Pseudo Multi-modal Features. *IEEE Transactions on Multimedia* (2024).

[114] Naomi Harte and Eoin Gillen. 2015. TCD-TIMIT: An audio-visual corpus of continuous speech. *IEEE Transactions on Multimedia* 17, 5 (2015), 603–615.

[115] Tianyu He, Junliang Guo, Runyi Yu, Yuchi Wang, Jialiang Zhu, Kaikai An, Leyi Li, Xu Tan, Chunyu Wang, Han Hu, et al. 2023. Gaia: Zero-shot talking avatar generation. *arXiv preprint arXiv:2311.15230* (2023).

[15] Martin Heusel, Hubert Ramsauer, Thomas Unterthiner, Bernhard Nessler, and Sepp Hochreiter. 2017. Gans trained by a two time-scale update rule converge to a local nash equilibrium. *Advances in neural information processing systems* 30 (2017).

[117] Timothy O Hodson. 2022. Root mean square error (RMSE) or mean absolute error (MAE): When to use them or not. *Geoscientific Model Development Discussions* 2022 (2022), 1–10.

[118] Fa-Ting Hong, Li Shen, and Dan Xu. 2023. DaGAN++: Depth-Aware Generative Adversarial Network for Talking Head Video Generation. arXiv:2305.06225 [cs.CV] https://arxiv.org/abs/2305.06225

[119] Fa-Ting Hong and Dan Xu. 2023. Implicit identity representation conditioned memory compensation network for talking head video generation. In *Proceedings of the IEEE/CVF international conference on computer vision*. 23062–23072.

[120] Fa-Ting Hong, Longhao Zhang, Li Shen, and Dan Xu. 2022. Depth-aware generative adversarial network for talking head video generation. In *Proceedings of the IEEE/CVF conference on computer vision and pattern recognition*. 3397–3406.

[16] Alain Hore and Djemel Ziou. 2010. Image quality metrics: PSNR vs. SSIM. In *2010 20th international conference on pattern recognition*. IEEE, 2366–2369.

[122] Youngjoon Jang, Kyeongha Rho, Jongbin Woo, Hyeongkeun Lee, Jihwan Park, Youshin Lim, Byeong-Yeol Kim, and Joon Son Chung. 2023. That's what i said: Fully-controllable talking face generation. In *Proceedings of the 31st ACM International Conference on Multimedia*. 3827–3836.

[123] Xiaozhong Ji, Xiaobin Hu, Zhihong Xu, Junwei Zhu, Chuming Lin, Qingdong He, Jiangning Zhang, Donghao Luo, Yi Chen, Qin Lin, et al. 2025. Sonic: Shifting focus to global audio perception in portrait animation. In *Proceedings of the Computer Vision and Pattern Recognition Conference*. 193–203.

[124] Xinya Ji, Hang Zhou, Kaisiyuan Wang, Wayne Wu, Chen Change Loy, Xun Cao, and Feng Xu. 2021. Audio-driven emotional video portraits. In *Proceedings of the IEEE/CVF conference on computer vision and pattern recognition*. 14080–14089.

[18] Diqiong Jiang, Jian Chang, Lihua You, Shaojun Bian, Robert Kosk, and Greg Maguire. 2024. Audio-Driven Facial Animation with Deep Learning: A Survey. *Information* 15, 11 (2024), 675.

[126] Jianwen Jiang, Gaojie Lin, Zhengkun Rong, Chao Liang, Yongming Zhu, Jiaqi Yang, and Tianyun Zhong. 2025. Mobileportrait: Real-time one-shot neural head avatars on mobile devices. In *Proceedings of the Computer Vision and Pattern Recognition Conference*. 15920–15929.

[127] Tero Karras, Timo Aila, Samuli Laine, and Jaakko Lehtinen. 2017. Progressive growing of gans for improved quality, stability, and variation. *arXiv preprint arXiv:1710.10196* (2017).

[128] Tero Karras, Samuli Laine, and Timo Aila. 2019. A style-based generator architecture for generative adversarial networks. In *Proceedings of the IEEE/CVF conference on computer vision and pattern recognition*. 4401–4410.

[19] Iqra Khan, Kashif Khan, and Arshad Ahmad. 2025. A Comprehensive Survey of DeepFake Generation and Detection Techniques in Audio-Visual Media. *ICCK Journal of Image Analysis and Processing* 1, 2 (2025), 73–95.

[130] Taekyung Ki and Dongchan Min. 2023. Stylelipsync: Style-based personalized lip-sync video generation. In *Proceedings of the IEEE/CVF international conference on computer vision*. 22841–22850.

[131] Taekyung Ki, Dongchan Min, and Gyeongsu Chae. 2025. Float: Generative motion latent flow matching for audio-driven talking portrait. In *Proceedings of the IEEE/CVF International Conference on Computer Vision*. 14699–14710.

[132] Jisoo Kim, Jungbin Cho, Joonho Park, Soonmin Hwang, Da Eun Kim, Geon Kim, and Youngjae Yu. 2025. DEEPTalk: Dynamic Emotion Embedding for Probabilistic Speech-Driven 3D Face Animation. In *Proceedings of the AAAI Conference on Artificial Intelligence*, Vol. 39. 4275–4283.

[20] Seyeon Kim, Siyoon Jin, Jihye Park, Kihong Kim, Jiyoung Kim, Jisu Nam, and Seungryong Kim. 2025. Moditalker: Motion-disentangled diffusion model for high-fidelity talking head generation. In *Proceedings of the AAAI Conference on Artificial Intelligence*, Vol. 39. 4302–4310.

[134] Avisek Lahiri, Vivek Kwatra, Christian Frueh, John Lewis, and Chris Bregler. 2021. Lipsync3d: Data-efficient learning of personalized 3d talking faces from video using pose and lighting normalization. In *Proceedings of the IEEE/CVF conference on computer vision and pattern recognition*. 2755–2764.

[135] Chunyu Li, Chao Zhang, Weikai Xu, Jinghui Xie, Weiguo Feng, Bingyue Peng, and Weiwei Xing. 2024. Latentsync: Audio conditioned latent diffusion models for lip sync. *arXiv e-prints* (2024), arXiv–2412.

[136] Dongze Li, Kang Zhao, Wei Wang, Bo Peng, Yingya Zhang, Jing Dong, and Tieniu Tan. 2024. Ae-nerf: Audio enhanced neural radiance field for few shot talking head synthesis. In *Proceedings of the AAAI Conference on Artificial Intelligence*, Vol. 38. 3037–3045.

[137] Hao Li, Ju Dai, Xin Zhao, Feng Zhou, Junjun Pan, and Lei Li. 2025. Wav2Sem: Plug-and-Play Audio Semantic Decoupling for 3D Speech-Driven Facial Animation. In *Proceedings of the Computer Vision and Pattern Recognition Conference*. 183–192.

[138] Jiahe Li, Jiawei Zhang, Xiao Bai, Jin Zheng, Xin Ning, Jun Zhou, and Lin Gu. 2024. Talkinggaussian: Structure-persistent 3d talking head synthesis via gaussian splatting. In *European Conference on Computer Vision*. Springer, 127–145.

[139] Jiahe Li, Jiawei Zhang, Xiao Bai, Jin Zheng, Jun Zhou, and Lin Gu. 2025. InsTaG: Learning Personalized 3D Talking Head from Few-Second Video. In *Proceedings of the Computer Vision and Pattern Recognition Conference*. 10690–10700.

[140] Jiahe Li, Jiawei Zhang, Xiao Bai, Jun Zhou, and Lin Gu. 2023. Efficient region-aware neural radiance fields for high-fidelity talking portrait synthesis. In *Proceedings of the IEEE/CVF International Conference on Computer Vision*. 7568–7578.

[22] Tianqi Li, Ruobing Zheng, Minghui Yang, Jingdong Chen, and Ming Yang. 2025. Ditto: Motion-space diffusion for controllable realtime talking head synthesis. In *Proceedings of the 33rd ACM International Conference on Multimedia*. 9704–9713.

[23] Weichuang Li, Longhao Zhang, Dong Wang, Bin Zhao, Zhigang Wang, Mulin Chen, Bang Zhang, Zhongjian Wang, Liefeng Bo, and Xuelong Li. 2023. One-shot high-fidelity talking-head synthesis with deformable neural radiance field. In *Proceedings of the IEEE/CVF Conference on Computer Vision and Pattern Recognition*. 17969–17978.

[143] Xuanchen Li, Jianyu Wang, Yuhao Cheng, Yikun Zeng, Xingyu Ren, Wenhan Zhu, Weiming Zhao, and Yichao Yan. 2025. Towards High-fidelity 3D Talking Avatar with Personalized Dynamic Texture. In *Proceedings of the Computer Vision and Pattern Recognition Conference*. 204–214.

[144] Yuan Li, Ziqian Bai, Feitong Tan, Zhaopeng Cui, Sean Fanello, and Yinda Zhang. 2025. IM-Portrait: Learning 3D-aware Video Diffusion for Photorealistic Talking Heads from Monocular VideosC. In *Proceedings of the Computer Vision and Pattern Recognition Conference*. 21107–21116.

[145] Yihong Lin, Zhaoxin Fan, Xianjia Wu, Lingyu Xiong, Liang Peng, Xiandong Li, Wenxiong Kang, Songju Lei, and Huang Xu. 2024. Glditalker: Speech-driven 3d facial animation with graph latent diffusion transformer. *arXiv preprint arXiv:2408.01826* (2024).

[146] Yukang Lin, Hokit Fung, Jianjin Xu, Zeping Ren, Adela SM Lau, Guosheng Yin, and Xiu Li. 2025. Mvportrait: Text-guided motion and emotion control for multi-view vivid portrait animation. In *Proceedings of the Computer Vision and Pattern Recognition Conference*. 26242–26252.

[147] Chang Liu, Ye Pan, Chenyang Ding, Susanto Rahardja, and Xiaokang Yang. 2025. Medtalk: Multimodal controlled 3d facial animation with dynamic emotions by disentangled embedding. In *Proceedings of the 33rd ACM International Conference on Multimedia*. 7538–7547.

[148] Huaize Liu, Wenzhang Sun, Donglin Di, Shibo Sun, Jiahui Yang, Changqing Zou, and Hujun Bao. 2025. Moee: Mixture of emotion experts for audio-driven portrait animation. In *Proceedings of the Computer Vision and Pattern Recognition Conference*. 26222–26231.

[24] Jiahe Liu, Youran Qu, Qi Yan, Xiaohui Zeng, Lele Wang, and Renjie Liao. 2024. Fr\'echet Video Motion Distance: A Metric for Evaluating Motion Consistency in Videos. *arXiv preprint arXiv:2407.16124* (2024).

[150] Ping Liu, Qiqi Tao, and Joey Tianyi Zhou. 2024. Evolving from single-modal to multi-modal facial deepfake detection: A survey. *arXiv e-prints* (2024), arXiv–2406.

[151] Tao Liu, Feilong Chen, Shuai Fan, Chenpeng Du, Qi Chen, Xie Chen, and Kai Yu. 2024. Anitalker: animate vivid and diverse talking faces through identity-decoupled facial motion encoding. In *Proceedings of the 32nd ACM International Conference on Multimedia*. 6696–6705.

[152] Tao Liu, Ziyang Ma, Qi Chen, Feilong Chen, Shuai Fan, Xie Chen, and Kai Yu. 2025. VQTalker: Towards Multilingual Talking Avatars Through Facial Motion Tokenization. In *Proceedings of the AAAI Conference on Artificial Intelligence*, Vol. 39. 5586–5594.

[153] Yunfei Liu, Lijian Lin, Fei Yu, Changyin Zhou, and Yu Li. 2023. Moda: Mapping-once audio-driven portrait animation with dual attentions. In *Proceedings of the IEEE/CVF International Conference on Computer Vision*. 23020–23029.

[154] Yujian Liu, Shidang Xu, Jing Guo, Dingbin Wang, Zairan Wang, Xianfeng Tan, and Xiaoli Liu. 2025. SyncAnimation: A Real-Time End-to-End Framework for Audio-Driven Human Pose and Talking Head Animation. *arXiv preprint arXiv:2501.14646* (2025).

[155] Ziwei Liu, Ping Luo, Xiaogang Wang, and Xiaoou Tang. 2015. Deep learning face attributes in the wild. In *Proceedings of the IEEE international conference on computer vision*. 3730–3738.

[25] Steven R Livingstone and Frank A Russo. 2018. The Ryerson Audio-Visual Database of Emotional Speech and Song (RAVDESS): A dynamic, multimodal set of facial and vocal expressions in North American English. *PloS one* 13, 5 (2018), e0196391.

[157] Yifeng Ma, Suzhen Wang, Yu Ding, Bowen Ma, Tangjie Lv, Changjie Fan, Zhipeng Hu, Zhidong Deng, and Xin Yu. 2025. Talkclip: Talking head generation with text-guided expressive speaking styles. *IEEE Transactions on Multimedia* (2025).

[158] Yifeng Ma, Suzhen Wang, Zhipeng Hu, Changjie Fan, Tangjie Lv, Yu Ding, Zhidong Deng, and Xin Yu. 2023. Styletalk: One-shot talking head generation with controllable speaking styles. In *Proceedings of the AAAI conference on artificial intelligence*, Vol. 37. 1896–1904.

[159] Moustafa Meshry, Saksham Suri, Larry S Davis, and Abhinav Shrivastava. 2021. Learned spatial representations for few-shot talking-head synthesis. In *Proceedings of the IEEE/CVF international conference on computer vision*. 13829–13838.

[160] Anish Mittal, Anush Krishna Moorthy, and Alan Conrad Bovik. 2012. No-reference image quality assessment in the spatial domain. *IEEE Transactions on image processing* 21, 12 (2012), 4695–4708.

[161] Anish Mittal, Rajiv Soundararajan, and Alan C Bovik. 2012. Making a "completely blind" image quality analyzer. *IEEE Signal processing letters* 20, 3 (2012), 209–212.

[162] Ali Mollahosseini, Behzad Hasani, and Mohammad H Mahoor. 2017. Affectnet: A database for facial expression, valence, and arousal computing in the wild. *IEEE Transactions on Affective Computing* 10, 1 (2017), 18–31.

[29] Muhammad Ferjad Naeem, Seong Joon Oh, Youngjung Uh, Yunjey Choi, and Jaejun Yoo. 2020. Reliable fidelity and diversity metrics for generative models. In *International conference on machine learning*. PMLR, 7176–7185.

[30] Arsha Nagrani, Joon Son Chung, and Andrew Zisserman. 2017. Voxceleb: a large-scale speaker identification dataset. *arXiv preprint arXiv:1706.08612* (2017).

[165] Niranjan D Narvekar and Lina J Karam. 2011. A no-reference image blur metric based on the cumulative probability of blur detection (CPBD). *IEEE Transactions on Image Processing* 20, 9 (2011), 2678–2683.

[166] OECD.AI. 2025. Frames Per Second (FPS) – AI performance metric. https://oecd.ai/en/catalogue/metrics/frames-per-second-fps.

[167] Vassil Panayotov, Guoguo Chen, Daniel Povey, and Sanjeev Khudanpur. 2015. Librispeech: an asr corpus based on public domain audio books. In *2015 IEEE international conference on acoustics, speech and signal processing (ICASSP)*. IEEE, 5206–5210.

[168] Se Jin Park, Minsu Kim, Joanna Hong, Jeongsoo Choi, and Yong Man Ro. 2022. Synctalkface: Talking face generation with precise lip-syncing via audio-lip memory. In *Proceedings of the AAAI Conference on Artificial Intelligence*, Vol. 36. 2062–2070.

[169] Ziqiao Peng, Yanbo Fan, Haoyu Wu, Xuan Wang, Hongyan Liu, Jun He, and Zhaoxin Fan. 2025. Dualtalk: Dual-speaker interaction for 3d talking head conversations. In *Proceedings of the Computer Vision and Pattern Recognition Conference*. 21055–21064.

[170] Ziqiao Peng, Wentao Hu, Junyuan Ma, Xiangyu Zhu, Xiaomei Zhang, Hao Zhao, Hui Tian, Jun He, Hongyan Liu, and Zhaoxin Fan. 2025. SyncTalk++: High-Fidelity and Efficient Synchronized Talking Heads Synthesis Using Gaussian Splatting. *arXiv preprint arXiv:2506.14742* (2025).

[171] Ziqiao Peng, Wentao Hu, Yue Shi, Xiangyu Zhu, Xiaomei Zhang, Hao Zhao, Jun He, Hongyan Liu, and Zhaoxin Fan. 2024. Synctalk: The devil is in the synchronization for talking head synthesis. In *Proceedings of the IEEE/CVF Conference on Computer Vision and Pattern Recognition*. 666–676.

[172] Ziqiao Peng, Yihao Luo, Yue Shi, Hao Xu, Xiangyu Zhu, Hongyan Liu, Jun He, and Zhaoxin Fan. 2023. Selftalk: A self-supervised commutative training diagram to comprehend 3d talking faces. In *Proceedings of the 31st ACM International Conference on Multimedia*. 5292–5301.

[173] Ziqiao Peng, Haoyu Wu, Zhenbo Song, Hao Xu, Xiangyu Zhu, Jun He, Hongyan Liu, and Zhaoxin Fan. 2023. Emotalk: Speech-driven emotional disentanglement for 3d face animation. In *Proceedings of the IEEE/CVF international conference on computer vision*. 20687–20697.

[33] KR Prajwal, Rudrabha Mukhopadhyay, Vinay P Namboodiri, and CV Jawahar. 2020. A lip sync expert is all you need for speech to lip generation in the wild. In *Proceedings of the 28th ACM international conference on multimedia*. 484–492.

[175] Feng Qiu, Wei Zhang, Chen Liu, Rudong An, Lincheng Li, Yu Ding, Changjie Fan, Zhipeng Hu, and Xin Yu. 2024. FreeAvatar: Robust 3D Facial Animation Transfer by Learning an Expression Foundation Model. In *SIGGRAPH Asia 2024 Conference Papers*. 1–11.

[176] Nabyl Quignon, Baptiste Chopin, Yaohui Wang, and Antitza Dantcheva. 2025. THEval. Evaluation Framework for Talking Head Video Generation. *arXiv preprint arXiv:2511.04520* (2025).

[35] Vineet Kumar Rakesh, Soumya Mazumdar, Research Pratim Maity, Sarbajit Pal, Amitabha Das, and Tapas Samanta. 2025. Advancing Talking Head Generation: A Comprehensive Survey of Multi-Modal Methodologies, Datasets, Evaluation Metrics, and Loss Functions. *arXiv preprint arXiv:2507.02900* (2025).

[178] Alexander Richard, Michael Zollhöfer, Yandong Wen, Fernando De la Torre, and Yaser Sheikh. 2021. Meshtalk: 3d face animation from speech using cross-modality disentanglement. In *Proceedings of the IEEE/CVF international conference on computer vision*. 1173–1182.

[179] Nicholas Ruiz and Marcello Federico. 2015. Phonetically-oriented word error alignment for speech recognition error analysis in speech translation. In *2015 IEEE Workshop on Automatic Speech Recognition and Understanding (ASRU)*. IEEE, 296–302.

[38] José Salas-Cáceres, Javier Lorenzo-Navarro, David Freire-Obregón, and Modesto Castrillón-Santana. 2025. Multimodal emotion recognition based on a fusion of audiovisual information with temporal dynamics. *Multimedia tools and applications* 84, 23 (2025), 27327–27343.

[40] Shuai Shen, Wenliang Zhao, Zibin Meng, Wanhua Li, Zheng Zhu, Jie Zhou, and Jiwen Lu. 2023. Difftalk: Crafting diffusion models for generalized audio-driven portraits animation. In *Proceedings of the IEEE/CVF conference on computer vision and pattern recognition*. 1982–1991.

[182] Changchong Sheng, Gangyao Kuang, Liang Bai, Chenping Hou, Yulan Guo, Xin Xu, Matti Pietikäinen, and Li Liu. 2024. Deep learning for visual speech analysis: A survey. *IEEE Transactions on Pattern Analysis and Machine Intelligence* 46, 9 (2024), 6001–6022.

[41] Zhaofeng Shi. 2021. A survey on audio synthesis and audio-visual multimodal processing. *arXiv preprint arXiv:2108.00443* (2021).

[184] Sonam Singh and Amol Dhumane. 2025. Unmasking digital deceptions: An integrative review of deepfake detection, multimedia forensics, and cybersecurity challenges. *MethodsX* (2025), 103632.

[42] Sanjana Sinha, Sandika Biswas, Ravindra Yadav, and Brojeshwar Bhowmick. 2022. Emotion-controllable generalized talking face generation. *arXiv preprint arXiv:2205.01155* (2022).

[43] Jake Snell, Karl Ridgeway, Renjie Liao, Brett D Roads, Michael C Mozer, and Richard S Zemel. 2017. Learning to generate images with perceptual similarity metrics. In *2017 IEEE international conference on image processing (ICIP)*. IEEE, 4277–4281.

[187] Stefan Stan, Kazi Injamamul Haque, and Zerrin Yumak. 2023. Facediffuser: Speech-driven 3d facial animation synthesis using diffusion. In *Proceedings of the 16th ACM SIGGRAPH Conference on Motion, Interaction and Games*. 1–11.

[188] Zhiyao Sun, Tian Lv, Sheng Ye, Matthieu Lin, Jenny Sheng, Yu-Hui Wen, Minjing Yu, and Yong-jin Liu. 2024. Diffposetalk: Speech-driven stylistic 3d facial animation and head pose generation via diffusion models. *ACM Transactions on Graphics (TOG)* 43, 4 (2024), 1–9.

[189] Shuai Tan, Bin Ji, and Ye Pan. 2024. Flowvqtalker: High-quality emotional talking face generation through normalizing flow and quantization. In *Proceedings of the IEEE/CVF Conference on Computer Vision and Pattern Recognition*. 26317–26327.

[190] Shuai Tan, Bin Ji, and Ye Pan. 2024. Style2talker: High-resolution talking head generation with emotion style and art style. In *Proceedings of the AAAI Conference on Artificial Intelligence*, Vol. 38. 5079–5087.

[191] Weipeng Tan, Chuming Lin, Chengming Xu, FeiFan Xu, Xiaobin Hu, Xiaozhong Ji, Junwei Zhu, Chengjie Wang, and Yanwei Fu. 2025. Disentangle Identity, Cooperate Emotion: Correlation-Aware Emotional Talking Portrait Generation. In *Proceedings of the 33rd ACM International Conference on Multimedia*. 9987–9995.

[192] Anni Tang, Tianyu He, Xu Tan, Jun Ling, Runnan Li, Sheng Zhao, Jiang Bian, and Li Song. 2024. Memories are one-to-many mapping alleviators in talking face generation. *IEEE Transactions on Pattern Analysis and Machine Intelligence* 46, 12 (2024), 8758–8770.

[193] Jiaxiang Tang, Kaisiyuan Wang, Hang Zhou, Xiaokang Chen, Dongliang He, Tianshu Hu, Jingtuo Liu, Ziwei Liu, Gang Zeng, and Jingdong Wang. 2025. Real-time neural radiance talking portrait synthesis via audio-spatial decomposition. *International Journal of Computer Vision* (2025), 1–12.

[194] Balamurugan Thambiraja, Ikhsanul Habibie, Sadegh Aliakbarian, Darren Cosker, Christian Theobalt, and Justus Thies. 2023. Imitator: Personalized speech-driven 3d facial animation. In *Proceedings of the IEEE/CVF international conference on computer vision*. 20621–20631.

[47] Mukhiddin Toshpulatov, Wookey Lee, and Suan Lee. 2023. Talking human face generation: A survey. *Expert Systems with Applications* 219 (2023), 119678.

[196] Gustave Udahemuka, Karim Djouani, and Anish M Kurien. 2024. Multimodal Emotion Recognition using visual, vocal and Physiological Signals: a review. *Applied Sciences* 14, 17 (2024), 8071.

[197] Thomas Unterthiner, Sjoerd Van Steenkiste, Karol Kurach, Raphaël Marinier, Marcin Michalski, and Sylvain Gelly. 2019. FVD: A new metric for video generation. (2019).

[198] Duomin Wang, Yu Deng, Zixin Yin, Heung-Yeung Shum, and Baoyuan Wang. 2023. Progressive disentangled representation learning for fine-grained controllable talking head synthesis. In *Proceedings of the IEEE/CVF Conference on Computer Vision and Pattern Recognition*. 17979–17989.

[199] Haotian Wang, Yuzhe Weng, Yueyan Li, Zilu Guo, Jun Du, Shutong Niu, Jiefeng Ma, Shan He, Xiaoyan Wu, Qiming Hu, et al. 2025. Emotivetalk: Expressive talking head generation through audio information decoupling and emotional video diffusion. In *Proceedings of the Computer Vision and Pattern Recognition Conference*. 26212–26221.

[200] Jiadong Wang, Xinyuan Qian, Malu Zhang, Robby T Tan, and Haizhou Li. 2023. Seeing what you said: Talking face generation guided by a lip reading expert. In *Proceedings of the IEEE/CVF Conference on Computer Vision and Pattern Recognition*. 14653–14662.

[50] Kaisiyuan Wang, Qianyi Wu, Linsen Song, Zhuoqian Yang, Wayne Wu, Chen Qian, Ran He, Yu Qiao, and Chen Change Loy. 2020. Mead: A large-scale audio-visual dataset for emotional talking-face generation. In *European conference on computer vision*. Springer, 700–717.

[202] Suzhen Wang, Lincheng Li, Yu Ding, Changjie Fan, and Xin Yu. 2021. Audio2head: Audio-driven one-shot talking-head generation with natural head motion. *arXiv preprint arXiv:2107.09293* (2021).

[203] Suzhen Wang, Lincheng Li, Yu Ding, and Xin Yu. 2022. One-shot talking face generation from single-speaker audio-visual correlation learning. In *Proceedings of the AAAI Conference on Artificial Intelligence*, Vol. 36. 2531–2539.

[204] Suzhen Wang, Yifeng Ma, Yu Ding, Zhipeng Hu, Changjie Fan, Tangjie Lv, Zhidong Deng, and Xin Yu. 2024. Styletalk++: A unified framework for controlling the speaking styles of talking heads. *IEEE Transactions on Pattern Analysis and Machine Intelligence* 46, 6 (2024), 4331–4347.

[205] Ting-Chun Wang, Arun Mallya, and Ming-Yu Liu. 2021. One-shot free-view neural talking-head synthesis for video conferencing. In *Proceedings of the IEEE/CVF conference on computer vision and pattern recognition*. 10039–10049.

[51] Zhou Wang, Alan C Bovik, Hamid R Sheikh, and Eero P Simoncelli. 2004. Image quality assessment: from error visibility to structural similarity. *IEEE transactions on image processing* 13, 4 (2004), 600–612.

[207] Zhou Wang, Eero P Simoncelli, and Alan C Bovik. 2003. Multiscale structural similarity for image quality assessment. In *The thrity-seventh asilomar conference on signals, systems & computers, 2003*, Vol. 2. Ieee, 1398–1402.

[208] Huawei Wei, Zejun Yang, and Zhisheng Wang. 2024. Aniportrait: Audio-driven synthesis of photorealistic portrait animation. *arXiv preprint arXiv:2403.17694* (2024).

[209] Rongliang Wu, Yingchen Yu, Fangneng Zhan, Jiahui Zhang, Xiaoqin Zhang, and Shijian Lu. 2023. Audio-driven talking face generation with diverse yet realistic facial animations. *Pattern Recognition* 144 (2023), 109865.

[210] Xiuzhe Wu, Pengfei Hu, Yang Wu, Xiaoyang Lyu, Yan-Pei Cao, Ying Shan, Wenming Yang, Zhongqian Sun, and Xiaojuan Qi. 2023. Speech2lip: High-fidelity speech to lip generation by learning from a short video. In *Proceedings of the IEEE/CVF international conference on computer vision*. 22168–22177.

[211] Liangbin Xie, Xintao Wang, Honglun Zhang, Chao Dong, and Ying Shan. 2022. Vfhq: A high-quality dataset and benchmark for video face super-resolution. In *Proceedings of the IEEE/CVF Conference on Computer Vision and Pattern Recognition*. 657–666.

[212] Yifan Xie, Tao Feng, Xin Zhang, Xiangyang Luo, Zixuan Guo, Weijiang Yu, Heng Chang, Fei Ma, and Fei Richard Yu. 2025. Pointtalk: Audio-driven dynamic lip point cloud for 3d gaussian-based talking head synthesis. In *Proceedings of the AAAI Conference on Artificial Intelligence*, Vol. 39. 8753–8761.

[53] Jinbo Xing, Menghan Xia, Yuechen Zhang, Xiaodong Cun, Jue Wang, and Tien-Tsin Wong. 2023. Codetalker: Speech-driven 3d facial animation with discrete motion prior. In *Proceedings of the IEEE/CVF Conference on Computer Vision and Pattern Recognition*. 12780–12790.

[54] Chao Xu, Yang Liu, Jiazhen Xing, Weida Wang, Mingze Sun, Jun Dan, Tianxin Huang, Siyuan Li, Zhi-Qi Cheng, Ying Tai, et al. 2024. Facechain-imagineid: Freely crafting high-fidelity diverse talking faces from disentangled audio. In *Proceedings of the IEEE/CVF Conference on Computer Vision and Pattern Recognition*. 1292–1302.

[215] Chao Xu, Junwei Zhu, Jiangning Zhang, Yue Han, Wenqing Chu, Ying Tai, Chengjie Wang, Zhifeng Xie, and Yong Liu. 2023. High-fidelity generalized emotional talking face generation with multi-modal emotion space learning. In *Proceedings of the IEEE/CVF conference on computer vision and pattern recognition*. 6609–6619.

[216] Zhihua Xu, Tianshui Chen, Zhijing Yang, Siyuan Peng, Keze Wang, and Liang Lin. 2025. Exploiting Temporal Audio-Visual Correlation Embedding for Audio-Driven One-Shot Talking Head Animation. *arXiv preprint arXiv:2504.05746* (2025).

[217] Zunnan Xu, Zhentao Yu, Zixiang Zhou, Jun Zhou, Xiaoyu Jin, Fa-Ting Hong, Xiaozhong Ji, Junwei Zhu, Chengfei Cai, Shiyu Tang, et al. 2025. Hunyuanportrait: Implicit condition control for enhanced portrait animation. In *Proceedings of the Computer Vision and Pattern Recognition Conference*. 15909–15919.

[218] Haijie Yang, Zhenyu Zhang, Hao Tang, Jianjun Qian, and Jian Yang. 2024. Consistentavatar: Learning to diffuse fully consistent talking head avatar with temporal guidance. In *Proceedings of the 32nd ACM International Conference on Multimedia*. 3964–3973.

[219] Ziyu Yao, Xuxin Cheng, and Zhiqi Huang. 2024. Fd2talk: Towards generalized talking head generation with facial decoupled diffusion model. In *Proceedings of the 32nd ACM International Conference on Multimedia*. 3411–3420.

[59] Zhenhui Ye, Ziyue Jiang, Yi Ren, Jinglin Liu, Jinzheng He, and Zhou Zhao. 2023. Geneface: Generalized and high-fidelity audio-driven 3d talking face synthesis. *arXiv preprint arXiv:2301.13430* (2023).

[221] Zipeng Ye, Mengfei Xia, Ran Yi, Juyong Zhang, Yu-Kun Lai, Xuwei Huang, Guoxin Zhang, and Yong-jin Liu. 2022. Audio-driven talking face video generation with dynamic convolution kernels. *IEEE Transactions on Multimedia* 25 (2022), 2033–2046.

[222] Zhenhui Ye, Tianyun Zhong, Yi Ren, Jiaqi Yang, Weichuang Li, Jiawei Huang, Ziyue Jiang, Jinzheng He, Rongjie Huang, Jinglin Liu, et al. 2024. Real3d-portrait: One-shot realistic 3d talking portrait synthesis. *arXiv preprint arXiv:2401.08503* (2024).

[223] Fei Yin, Yong Zhang, Xiaodong Cun, Mingdeng Cao, Yanbo Fan, Xuan Wang, Qingyan Bai, Baoyuan Wu, Jue Wang, and Yujiu Yang. 2022. Styleheat: One-shot high-resolution editable talking face generation via pre-trained stylegan. In *European conference on computer vision*. Springer, 85–101.

[61] Lijun Yin, Xiaozhou Wei, Yi Sun, Jun Wang, and Matthew J Rosato. 2006. A 3D facial expression database for facial behavior research. In *7th international conference on automatic face and gesture recognition (FGR06)*. IEEE, 211–216.

[225] Hongyun Yu, Zhan Qu, Qihang Yu, Jianchuan Chen, Zhonghua Jiang, Zhiwen Chen, Shengyu Zhang, Jimin Xu, Fei Wu, Chengfei Lv, et al. 2024. Gaussiantalker: Speaker-specific talking head synthesis via 3d gaussian splatting. In *Proceedings of the 32nd ACM International Conference on Multimedia*. 3548–3557.

[226] Jianhui Yu, Hao Zhu, Liming Jiang, Chen Change Loy, Weidong Cai, and Wayne Wu. 2023. Celebv-text: A large-scale facial text-video dataset. In *Proceedings of the IEEE/CVF Conference on Computer Vision and Pattern Recognition*. 14805–14814.

[227] Yu Yu, Weibin Zhang, and Yun Deng. 2021. Frechet inception distance (fid) for evaluating gans. *China University of Mining Technology Beijing Graduate School* 3, 11 (2021).

[228] Zhentao Yu, Zixin Yin, Deyu Zhou, Duomin Wang, Finn Wong, and Baoyuan Wang. 2023. Talking head generation with probabilistic audio-to-visual diffusion priors. In *Proceedings of the IEEE/CVF International Conference on Computer Vision*. 7645–7655.

[229] Bohan Zeng, Xuhui Liu, Sicheng Gao, Boyu Liu, Hong Li, Jianzhuang Liu, and Baochang Zhang. 2023. Face animation with an attribute-guided diffusion model. In *Proceedings of the IEEE/CVF Conference on Computer Vision and Pattern Recognition*. 628–637.

[62] Bowen Zhang, Chenyang Qi, Pan Zhang, Bo Zhang, HsiangTao Wu, Dong Chen, Qifeng Chen, Yong Wang, and Fang Wen. 2023. Metaportrait: Identity-preserving talking head generation with fast personalized adaptation. In *Proceedings of the IEEE/CVF Conference on Computer Vision and Pattern Recognition*. 22096–22105.

[231] Chenxu Zhang, Yifan Zhao, Yifei Huang, Ming Zeng, Saifeng Ni, Madhukar Budagavi, and Xiaohu Guo. 2021. Facial: Synthesizing dynamic talking face with implicit attribute learning. In *Proceedings of the IEEE/CVF international conference on computer vision*. 3867–3876.

[232] Haiming Zhang, Zhihao Yuan, Chaoda Zheng, Xu Yan, Baoyuan Wang, Guanbin Li, Song Wu, Shuguang Cui, and Zhen Li. 2025. Gsmoothface: Generalized smooth talking face generation via fine grained 3d face guidance. *IEEE Transactions on Visualization and Computer Graphics* (2025).

[233] Shiqing Zhang, Yijiao Yang, Chen Chen, Xingnan Zhang, Qingming Leng, and Xiaoming Zhao. 2024. Deep learning-based multimodal emotion recognition from audio, visual, and text modalities: A systematic review of recent advancements and future prospects. *Expert Systems with Applications* 237 (2024), 121692.

[63] Wenxuan Zhang, Xiaodong Cun, Xuan Wang, Yong Zhang, Xi Shen, Yu Guo, Ying Shan, and Fei Wang. 2023. Sadtalker: Learning realistic 3d motion coefficients for stylized audio-driven single image talking face animation. In *Proceedings of the IEEE/CVF conference on computer vision and pattern recognition*. 8652–8661.

[235] Yue Zhang, LIU Minhao, Zhaokang Chen, Bin Wu, Chao Zhan, Yingjie He, JUNXIN HUANG, Wenjiang Zhou, et al. 2024. Musetalk: Real-time high quality lip synchronization with latent space inpainting. (2024).

[236] Zhimeng Zhang, Zhipeng Hu, Wenjin Deng, Changjie Fan, Tangjie Lv, and Yu Ding. 2023. Dinet: Deformation inpainting network for realistic face visually dubbing on high resolution video. In *Proceedings of the AAAI conference on artificial intelligence*, Vol. 37. 3543–3551.

[64] Zhimeng Zhang, Lincheng Li, Yu Ding, and Changjie Fan. 2021. Flow-guided one-shot talking face generation with a high-resolution audio-visual dataset. In *Proceedings of the IEEE/CVF conference on computer vision and pattern recognition*. 3661–3670.

[238] Zhicheng Zhang, Yibo Sun, and Shiyan Su. 2023. Multimodal learning for automatic summarization: a survey. In *International Conference on Advanced Data Mining and Applications*. Springer, 362–376.

[239] Shuling Zhao, Fa-Ting Hong, Xiaoshui Huang, and Dan Xu. 2025. Synergizing motion and appearance: Multi-scale compensatory codebooks for talking head video generation. In *Proceedings of the Computer Vision and Pattern Recognition Conference*. 26232–26241.

[240] Dingcheng Zhen, Shunshun Yin, Shiyang Qin, Hou Yi, Ziwei Zhang, Siyuan Liu, Gan Qi, and Ming Tao. 2025. Teller: Real-Time Streaming Audio-Driven Portrait Animation with Autoregressive Motion Generation. In *Proceedings of the Computer Vision and Pattern Recognition Conference*. 21075–21085.

[65] Le Zheng, Hu Yongting, and Xu Yong. 2025. Survey of Audio-Driven Talking Face Video Generation and Identification. *Journal of Computer Research and Development* 62, 10 (2025), 2523–2544. doi:10.7544/issn1000-1239.202440207

[242] Tianyun Zhong, Chao Liang, Jianwen Jiang, Gaojie Lin, Jiaqi Yang, and Zhou Zhao. 2025. Fada: Fast diffusion avatar synthesis with mixed-supervised multi-cfg distillation. In *Proceedings of the Computer Vision and Pattern Recognition Conference*. 3101–3110.

[243] Weizhi Zhong, Chaowei Fang, Yinqi Cai, Pengxu Wei, Gangming Zhao, Liang Lin, and Guanbin Li. 2023. Identity-preserving talking face generation with landmark and appearance priors. In *Proceedings of the IEEE/CVF Conference on Computer Vision and Pattern Recognition*. 9729–9738.

[244] Hang Zhou, Yasheng Sun, Wayne Wu, Chen Change Loy, Xiaogang Wang, and Ziwei Liu. 2021. Pose-controllable talking face generation by implicitly modularized audio-visual representation. In *Proceedings of the IEEE/CVF conference on computer vision and pattern recognition*. 4176–4186.

[245] Hao Zhu, Man-Di Luo, Rui Wang, Ai-Hua Zheng, and Ran He. 2021. Deep audio-visual learning: A survey. *International Journal of Automation and Computing* 18, 3 (2021), 351–376.

[67] Hao Zhu, Wayne Wu, Wentao Zhu, Liming Jiang, Siwei Tang, Li Zhang, Ziwei Liu, and Chen Change Loy. 2022. CelebV-HQ: A large-scale video facial attributes dataset. In *European conference on computer vision*. Springer, 650–667.

[247] Yixiang Zhuang, Baoping Cheng, Yao Cheng, Yuntao Jin, Renshuai Liu, Chengyang Li, Xuan Cheng, Jing Liao, and Juncong Lin. 2024. Learn2talk: 3d talking face learns from 2d talking face. *IEEE Transactions on Visualization and Computer Graphics* (2024).