# OpenReview forum: "Talking-Head Generation in Practice"
_ACM.org/TheWebConf/2026/Workshop/TIME — TIME 2026 Oral_

### Official Review · Reviewer_zrwZ · 2025-12-30
**Detailed survey paper with good structure**

**Rating:** 7
**Confidence:** 4

**Review:**

The paper presents a survey and analysis of the "Talking-Head Generation", covering 117 papers (2021-2025). The authors aim to restructure the field’s understanding by proposing a semantic taxonomy, quantitative analysis of dataset usage, and rise & fall of evaluation metrics. They highlight how trends shifted (from frame-level to synchronization, emotion, controllability). Additionally, they provided short summary under each subsection and also discussed ethical considerations towards the end.

- Figure 1 clustering (TF-IDF) can be confusing, as it mixes architectures (diffusion models, NeRF based..) with goals ("emotion driven", "style controllable"). What is X axis in Figure 1?
- What is Y axis in Figure 2?
- Paper used "LLM-based extraction prompt" to extract data from papers. Since LLMs can hallucinate, how (if) the results were validated?
- In “style- and emotion-aware”, “-” should be removed

---

### Official Review · Reviewer_kKu8 · 2026-01-03
**This paper presents a large-scale empirical survey of talking-head generation research from 2021 to 2025, analyzing 117 papers through dataset usage trends, metric evolution, and a data-driven taxonomy of methods. The work provides a valuable overview of how evaluation practices and modeling paradigms have evolved, with practical recommendations aimed at improving reproducibility and evaluation consistency in the field.**

**Rating:** 7
**Confidence:** 4

**Review:**

### Evaluation of Quality, Clarity, Originality, and Significance

This paper presents a large-scale empirical survey of talking-head generation research, analyzing 117 papers published between 2021 and 2025. Overall, the work is **high quality, clearly written, and methodologically sound**, offering a structured and data-driven perspective on how datasets, evaluation metrics, and modeling paradigms have evolved in this rapidly growing field. While the contribution is primarily analytical rather than technical, the paper provides **clear practical value**, particularly with respect to evaluation practices and reproducibility.

### Strengths

**Comprehensive Empirical Coverage**
The analysis spans a substantial corpus of 117 papers, providing **longitudinal, quantitative evidence** of shifts in dataset usage and evaluation metrics. The figures summarizing dataset and metric trends effectively ground the discussion in data rather than anecdotal observations.

**Data-Driven Taxonomy Construction**
The use of TF-IDF representations and hierarchical clustering to derive a taxonomy offers a **more objective alternative** to manually curated categorizations used in prior surveys. The resulting clusters meaningfully capture convergence between emotion-driven synthesis, controllable animation, and audio-visual alignment approaches.

**Practical Evaluation Guidance**
Section 3.4 is a strong component of the paper, translating observed trends into **actionable recommendations** for evaluation protocols. The proposed compact metric suite balancing perceptual, temporal, geometric, and controllability signals is well motivated and likely to be useful for practitioners and benchmark designers.

**Well-Documented Reference Resource**
The appendix tables summarizing datasets, metrics, and implementation details across the reviewed papers constitute a **valuable reference** for the community. Table 1 provides a concise and accessible overview that enhances the paper’s usability as a survey.

**Ethical Awareness**
The discussion of ethical considerations, including misuse risks, consent, and demographic bias, is appropriate and timely given the sensitive nature of talking-head generation.

### Weaknesses (Evaluation-Focused)

**Limited Technical Novelty**
The paper is primarily a survey and empirical analysis and does not introduce new models or theoretical frameworks.

**Missing Metric–Human Alignment Analysis**
The paper does not analyze relationships between automated metrics and human judgments (e.g., MOS). A lightweight correlation analysis (e.g., Spearman’s $\rho$) would strengthen the evaluation recommendations.

**Taxonomy Robustness Not Examined**
The stability of the proposed taxonomy under alternative representations or clustering methods is not evaluated.

### Minor Observations

- Figure 1 is visually dense and could benefit from improved readability
- Validation details for LLM-based metadata extraction could be clarified

### Questions for the Authors

1. How robust is the proposed taxonomy to alternative document representations or clustering methods?
2. Were the LLM-based metadata extractions manually validated on a subset of papers?
3. Have you considered analyzing metric–MOS correlations using existing benchmarks?

### Suggestions for Improvement

- Add a lightweight metric–MOS correlation analysis where data is available
- Include basic robustness checks for the taxonomy
- Clarify validation procedures for automated metadata extraction

### Overall Assessment

This is a **well-executed and clearly written empirical survey** that makes a solid contribution by systematically documenting trends in talking-head generation and evaluation practices. While its originality is limited by its survey nature, the paper’s empirical rigor, clarity, and practical guidance make it a **valuable contribution** to the workshop.

---

### Official Review · Reviewer_s99F · 2026-01-07
**The paper is recommended for **Strong Accept** and serves as a vital **longitudinal survey** of the talking-head generation field from 2021 to 2025. Based on a corpus of **117 works**, it reveals a community-wide shift from achieving low-level pixel realism toward **semantic alignment, temporal coherence, and expressive controllability**.**

**Rating:** 9
**Confidence:** 4

**Review:**

#### **1. Quality**
The quality of this work is underpinned by a **rigorous and data-driven methodology** that distinguishes it from traditional, manually curated surveys. The authors processed a corpus of **117 representative papers** (2021–2025) using structured LLM-based extraction to normalise dataset names, metrics, and training/testing protocols. By employing **TF-IDF representations** and **hierarchical agglomerative clustering with Ward’s linkage**, the paper ensures that its taxonomy is a faithful reflection of the literature's organic structure rather than an imposed framework. The inclusion of GitHub popularity (star counts) and peer-review status as filtering signals further enhances the reliability of the selected works.

#### **2. Clarity**
The paper exhibits **high clarity** through a systematic organisation that guides the reader from technical foundations to practical benchmarking. Complex trends are made accessible via **visual analytics**, such as radar plots for metric evolution (Figure 4) and longitudinal bar charts for dataset usage (Figure 2). Furthermore, the authors define each evaluation metric in detail, providing mathematical formulations for clarity, such as the **Peak Signal-to-Noise Ratio (PSNR)**:
$$PSNR = 10 \log_{10} \left( \frac{MAX^2}{MSE} \right)$$
where $MSE$ is the Mean Squared Error. Similarly, the **Word Error Rate (WER)** is clearly defined as:
$$WER = \frac{S + D + I}{N}$$
where $S, D,$ and $I$ represent substitutions, deletions, and insertions.

#### **3. Originality**
The work is highly original as it presents the **first longitudinal analysis** of datasets and metrics specifically for the 2021–2025 period. While previous surveys focused on narrow subtasks like lip-sync or face reenactment, this paper maps the entire landscape of **expressive talking-head generation**. Its "data-driven" approach to taxonomy construction—bridging quantitative text analysis with LLM-assisted labelling—represents a novel meta-research technique in the field of computer vision.

#### **4. Significance**
This survey serves as a **central reference** for advancing expressive and controllable digital humans. Its significance lies in identifying a profound community shift from pixel-level fidelity (e.g., SSIM, PSNR) toward **semantic and behavioural alignment** (e.g., LSE-C/D, F-LMD). By providing **actionable guidelines**—including a recommended compact metric suite and specific test-split protocols for generalization—the paper offers a concrete roadmap for future benchmarking and reproducible research.

---

### **Pros and Cons**

#### **Pros**
*   **Comprehensive Taxonomy:** Successfully identifies ten coherent research directions, ranging from NeRF-based 3D heads to emotion-driven synthesis.
*   **Trend Identification:** Captures the decline of traditional metrics and the rise of **semantically oriented measures** like audio-visual synchronization.
*   **Dataset Mapping:** Documents the transition from generic benchmarks (VoxCeleb) to **high-resolution, emotion-rich, and multi-view datasets** like HDTF, MEAD, and RAVDESS.
*   **Benchmarking Guidance:** Proposes a standardised evaluation suite including identity similarity (ArcFace), landmark accuracy (LMD), and human Mean Opinion Score (MOS).
*   **Performance Metrics:** Includes practical runtime metrics such as **Frames Per Second (FPS)**:
    $$FPS = \frac{\text{number of output frames}}{\text{wall-clock time (seconds)}}$$
    essential for real-time applications.

#### **Cons**
*   **Methodological Sensitivity:** The clustering results depend on specific vocabulary choices in abstracts and the inherent priors of the LLMs used for labelling.
*   **Scope Limitations:** The survey is restricted to publicly available research and may miss **proprietary industrial practices** or unpublished benchmarks.
*   **Reporting Inconsistencies:** The extraction process is subject to how original papers reported their experiments, which can lead to occasional omissions.
*   **Emerging Ethical Standards:** While it discusses deepfakes and demographic bias, concrete **misuse-resilience criteria** for evaluation are still noted as an emerging, unstandardised area.

***

To further strengthen the research presented in the sources, one could pursue several lines of inquiry regarding methodological sensitivity, the inclusion of proprietary data, and the standardisation of emerging metrics.


### **Requests for Additional Data**
*   **Demographic Granularity:** The sources note that **demographic imbalances** can lead to unequal performance across gender, age, or ethnic groups. More data is needed on the specific **demographic composition** of the 117-paper corpus to understand if certain populations are systematically under-represented in current benchmarking.
*   **Human Evaluation Metadata:** While the sources recommend a **Mean Opinion Score (MOS)** with at least 30 raters, they do not provide the raw inter-rater reliability data from the surveyed papers. More data on how **subjective judgments** correlate with new semantic metrics like **LSE-C/D** would be valuable, as existing studies suggest this correlation is limited.
*   **Low-Shot Efficiency:** There is a request for more data on **"low-shot personalization"** (using only 1–5 frames) to determine the current state-of-the-art for identity stability when data cost is extremely low.

### **Proposed Improvements for Future Versions**
*   **Standardised Reporting Protocols:** To address **reporting inconsistencies** found in the experimental sections of the surveyed papers, the authors could propose a mandatory **"standardised reporting template"** for future talking-head publications.
*   **Misuse-Resilience Benchmarking:** While ethical risks are discussed, the work would be improved by including a concrete **"misuse-resilience" evaluation category**, which specifically tests a model's susceptibility to deepfake creation or its ability to support **watermarking and provenance signals**.
*   **Integration of Body Dynamics:** Future iterations should expand beyond the "head" to include **unified generative models** that integrate gaze behavior, emotional state, and **upper-body gesturing** into a single cohesive model, as this is noted as a long-term frontier.
*   **Semantic Grounding:** An improvement could be made by introducing more rigorous **linguistic context metrics** to evaluate if generated motions are contextually appropriate for the specific dialogue being spoken, rather than just being "synced" to the audio.

***


**Analogy for Understanding:**
Think of the talking-head generation field as a developing orchestra. Early research (2021) was focused on making sure each instrument could make a clear sound (pixel-level realism). This paper shows that the modern field (2025) has moved on to the "symphony" stage, where the focus is now on the timing, emotional expression, and how well the different sections (audio, motion, and identity) play together in perfect harmony (semantic and temporal alignment).

---

### Author Rebuttal · Authors · 2026-01-12

## Reviewer s99F

### Cons

**Methodological Sensitivity:**
The clustering results depend on specific vocabulary choices in abstracts and the inherent priors of the LLMs used for labeling.

**Response:**
We fully agree with the reviewer’s assessment. Semantic clustering is inherently sensitive to text representations and vocabulary distributions. In our work, the clustering structure is entirely determined by TF-IDF representations and Ward linkage, while the LLM is used only for post hoc label interpretation rather than for generating the cluster structure itself.

---

**Scope Limitations:**
The survey is restricted to publicly available research and may miss proprietary industrial practices or unpublished benchmarks.

**Response:**
We acknowledge this limitation and have further clarified in the revised manuscript that our survey focuses on *reproducible and publicly accessible research practices*. While industrial systems and private datasets are undoubtedly important, their lack of public verifiability makes them difficult to include in a systematic longitudinal analysis.

---

**Reporting Inconsistencies:**
The extraction process is subject to how original papers reported their experiments, which can lead to occasional omissions.

**Response:**
We agree that this issue reflects a broader lack of standardization in experimental reporting within the field. In practice, identical evaluation metrics are often referred to using different names, and experimental settings are described with varying levels of detail, which increases the complexity of information extraction and normalization and may lead to occasional omissions. In this study, we adopted a combination of unified mapping rules and manual verification to ensure a reasonable and robust extraction strategy and to mitigate these effects as much as possible.
At the same time, we appreciate the reviewer’s constructive suggestion and believe that addressing this issue ultimately requires community-level efforts. Accordingly, we highlight in the future outlook the potential value of introducing a standardized experimental reporting template to improve the completeness and reproducibility of survey-based analyses and benchmarking.

---

**Emerging Ethical Standards:**
While the paper discusses deepfakes and demographic bias, concrete misuse-resilience criteria for evaluation remain an emerging and unstandardized area.

**Response:**
We fully agree with the reviewer’s observation. Current discussions on deepfake risks, bias amplification, and content provenance are still at an exploratory stage, and no unified evaluation standards have yet emerged. The goal of this paper is not to propose a mature ethical evaluation framework, but rather to clearly identify misuse resilience as an important and currently unstandardized research frontier. By systematically reviewing existing discussions, we aim to help the community better recognize both the importance and the complexity of this issue.



We appreciate the reviewer’s thoughtful suggestions and agree that these directions represent important next milestones for the field.

In particular, demographic granularity and human evaluation metadata remain underreported across existing benchmarks, limiting large-scale longitudinal analysis. As reporting practices evolve, we plan to incorporate these dimensions into future versions of this survey. We are also actively exploring low-shot personalization settings in ongoing work, which we see as a key challenge for identity stability under minimal data regimes.

With respect to future improvements, we strongly support the call for standardized reporting templates and misuse-resilience benchmarking. We believe these will be essential for responsible deployment of expressive talking-head systems. Extending the scope toward unified head–body dynamics and richer semantic grounding is also a natural long-term direction that we intend to pursue.

We thank the reviewer for articulating these forward-looking considerations, which will help guide both our future work and broader community efforts.**We have further elaborated on all of the above points in Section E (Future Directions and Limitations) of the Appendix.**





---

## Reviewer kKu8

### Questions for the Authors

**Q1. How robust is the proposed taxonomy to alternative document representations or clustering methods?**

**Response:**
We believe that the proposed taxonomy is robust at the level of major research directions. The goal of this work is not to provide a unique or optimal categorization, but to reveal dominant research themes and their relationships through a unified, data-driven process. While alternative text representations or clustering strategies (e.g., different feature granularities or similarity measures) may affect the assignment of some boundary cases, such variations are expected to occur primarily between neighboring sub-directions rather than altering the overall structure of core themes such as audio-driven modeling, emotion and style control, and 3D or diffusion-based approaches. Given our focus on long-term trends and thematic convergence, a systematic robustness analysis across alternative representations is identified as a valuable direction for future work.

---

**Q2. Were the LLM-based metadata extractions manually validated on a subset of papers?**

**Response:**
Yes. In our pipeline, LLMs are used to assist with structured information extraction and normalization, rather than serving as fully automated decision-making components. In practice, extracted metadata were combined with rule-based constraints and manually verified, with particular attention to key fields such as dataset names, evaluation metrics, and training/testing splits. This design aims to balance scalability and consistency while improving the reliability and reproducibility of the extracted information.

---

**Q3. Have you considered analyzing metric–MOS correlations using existing benchmarks?**

**Response:**
We have reviewed the literature and note that while some works propose learned preference scores or MOS prediction models in related generative tasks, there is no commonly adopted talking-head benchmark that provides per-sample MOS with automatic metric scores. Most reported MOS values are aggregated, making large-scale correlation analysis infeasible. We therefore view systematic metric–MOS correlation as an important open problem and a recommendation for future standardized evaluation efforts.

---

## Reviewer zrwZ

**Comment:**
Figure 1 clustering (TF-IDF) can be confusing, as it mixes architectures (e.g., diffusion models, NeRF-based methods) with goals (e.g., emotion-driven, style-controllable). What is the X-axis in Figure 1?

**Response:**
Figure 1 presents a data-driven semantic hierarchy derived from TF-IDF representations of paper titles and abstracts. The x-axis denotes the linkage distance in hierarchical clustering using Ward’s method, which reflects semantic dissimilarity between groups of papers instead of a physical or methodological dimension. **We have revised the caption of Figure 1 to explicitly clarify that the x-axis represents the linkage distance in hierarchical clustering, thereby avoiding potential confusion.**

---

**Comment:**
What is the Y-axis in Figure 2?

**Response:**

The y-axis denotes the number of papers adopting a given evaluation metric or datasets in each year. **We have updated the captions of Figures 2 and 3 to explicitly clarify this point in the main text.**

---

**Comment:**
The paper uses an “LLM-based extraction prompt” to extract data from papers. Since LLMs can hallucinate, how (if at all) were the results validated?

**Response:**
Yes. LLMs are used only to assist with structured extraction and normalization, not as fully automated sources of ground truth. In practice, extracted results were combined with rule-based constraints and manually verified, with particular attention to dataset names, evaluation metrics, and training/testing protocols, in order to reduce errors caused by phrasing variations or naming inconsistencies.


**Comment:**
In “style- and emotion-aware,” the hyphen should be removed.

**Response:**
Thank you for the helpful suggestion. The hyphenated form *“style- and emotion-aware animation”* is intentionally used, following standard academic conventions for coordinated compound modifiers (e.g., *“audio- and video-driven”*). Here, *“style-aware”* and *“emotion-aware”* jointly modify *“animation”*, and the hyphenation helps avoid potential ambiguity and improves clarity.

---

### Meta-Review · Area_Chair_sMY1 · 2026-01-15

**Recommendation:** Accept (Oral)
**Confidence:** 5

**Metareview:**

This paper presents a well-executed and timely longitudinal survey of talking-head generation research, offering clear value to the community through its data-driven taxonomy and analysis of evaluation trends. The reviewers are aligned in their positive assessment, with critiques focused mainly on expected limitations of survey-based work rather than substantive flaws. The authors’ rebuttal is thorough and appropriately addresses concerns regarding methodology, validation, and scope. Overall, the paper is solid, clearly written, and suitable for acceptance. Given its breadth, maturity, and reference value, I recommend acceptance as an oral presentation.

---

### Decision · Program_Chairs · 2026-01-16

Accept (Oral)